# $\nabla^2$DFT: A Universal Quantum Chemistry Dataset of Drug-Like Molecules and a Benchmark for Neural Network Potentials

**Kuzma Khrabrov**[1, ✉]**, Anton Ber**[1]**, Artem Tsypin**[1]**, Konstantin Ushenin**[1]**, Egor Rumiantsev**[2]**,
Alexander Telepov**[1]**, Dmitry Protasov**[1]**, Ilya Shenbin**[6]**, Anton Alekseev**[3, 6]**, Mikhail Shirokikh**[3]**,
Sergey Nikolenko**[4, 6]**, Elena Tutubalina**[1, 4, 5]**, and Artur Kadurin**[1, 4, ✉]

[1]AIRI, Moscow
[2]EPFL, Lausanne
[3]St. Petersburg State University, St. Petersburg
[4]ISP RAS Research Center for Trusted Artificial Intelligence, Moscow
[5]Sber AI, Moscow
[6]St. Petersburg Department of the Steklov Institute of Mathematics, St. Petersburg
✉{Khrabrov, Kadurin}@airi.net

## Abstract

Methods of computational quantum chemistry provide accurate approximations of molecular properties crucial for computer-aided drug discovery and other areas of chemical science. However, high computational complexity limits the scalability of their applications. Neural network potentials (NNPs) are a promising alternative to quantum chemistry methods, but they require large and diverse datasets for training. This work presents a new dataset and benchmark called $\nabla^2$DFT that is based on the nablaDFT. It contains twice as much molecular structures, three times more conformations, new data types and tasks, and state-of-the-art models. The dataset includes energies, forces, 17 molecular properties, Hamiltonian and overlap matrices, and a wavefunction object. All calculations were performed at the DFT level ($\omega$B97X-D/def2-SVP) for each conformation. Moreover, $\nabla^2$DFT is the first dataset that contains relaxation trajectories for a substantial number of drug-like molecules. We also introduce a novel benchmark for evaluating NNPs in molecular property prediction, Hamiltonian prediction, and conformational optimization tasks. Finally, we propose an extendable framework for training NNPs and implement 10 models within it.

## 1 Introduction

Solving the many-particle Schrödniger equation (SE) for electrons makes it possible to describe the electronic structure of matter. This structure determines the equilibrium and transport properties of matter that are crucial in downstream applications such as computer-aided drug design or material design [1–8]. However, since an analytic solution to the many-particle SE is unknown, various approximate solutions are used in practice, leading to a trade-off between accuracy and computational cost. The most accurate methods, such as Post-Hartree-Fock [9] and quantum Monte Carlo methods [10] are prohibitively expensive, applicable to systems with at most tens of atoms; for a comprehensive overview of numerical methods at different levels of accuracy see [11].

*Density functional theory* (DFT) [12–14] is currently the primary approach for solving the many-particle SE for electrons. It provides reasonably accurate predictions while being computationally tractable for systems on a scale of 1000 electrons [15]. However, even a single iteration of this method

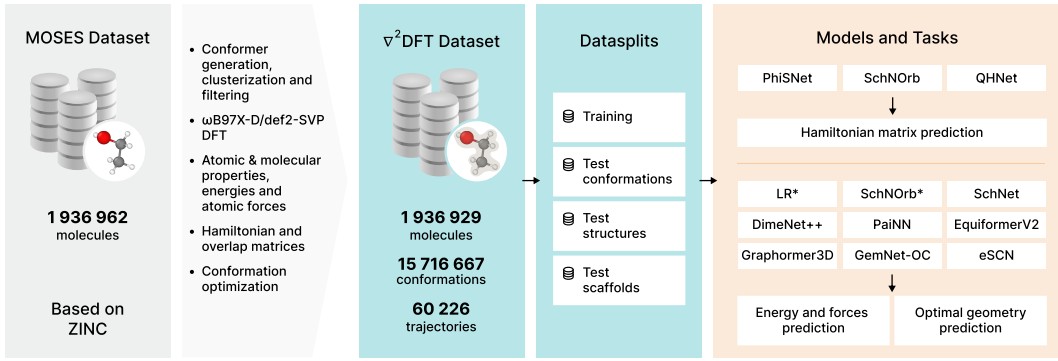

Figure 1: Our comprehensive workflow for dataset and benchmark construction as elaborated in Sections 3 and 4. First, a diverse set of conformations is generated for molecules from the MOSES dataset. Second, Quantum Chemistry (QC) properties are computed for these conformations, accompanied by optimization trajectories. Third, this data is then arranged into training and testing splits. Finally, ten state-of-the-art models are trained and evaluated based on these splits.

may take several CPU-hours [16], which restricts its use in molecular modeling tasks where a single method has to be called many times (e.g., more than $10^6$ times for molecular dynamics simulation). Neural networks have recently emerged as an alternative to the computationally expensive quantum chemistry (QC) approaches. However, they require substantial amounts of data for training. Since collecting data at the highest level of theory (such as Post-Hartree-Fock or Quantum Monte Carlo) is extremely expensive, most existing datasets (see Section 2) use DFT-based methods.

There are multiple ways to parametrize the solution of the SE equation with a neural network (NN). The most general approach is to directly predict the wavefunction of the atomic system [17–24], which allows to infer interesting properties of the system quickly. In general, this approach does not require a dataset to train on and does not depend on a specific method of solving the many-particle SE, but this family of methods is very resource-demanding [19]. Another approach is to predict the quantum Hamiltonian matrix [25–30], which fully defines the wavefunction on the Kohn-Sham density functional and Hartree-Fock levels of theory. This approach is not as general since it can only operate on certain levels of theory but retains all merits of wavefunction prediction. Finally, the most popular approach involves training *neural network potentials* (NNPs) [31–40], i.e., neural networks designed to predict potential energy and interatomic forces in atomic systems based on the structural arrangement of the atoms. Their inference time scales at most quadratically in the number of atoms in the system, which makes NNPs relatively cheap to train and applicable in such important tasks as molecular dynamics simulations [32, 41–43] and molecular conformation optimization [44–46].

In this work, we aim to aid the training of Hamiltonian-predicting models and NNPs for druglike molecules by significantly extending and improving the nablaDFT dataset [11]. We double the number of molecules and conformations to 1,936,929 and 12,676,264, respectively. We call the proposed dataset $\nabla^2$DFT. For each conformation, we provide various QC properties, including energy, forces, Hamiltonian and overlap matrices, and the wavefunction object that allows to either directly infer or calculate additional QC properties. All calculations were performed at the $\omega$B97X-D/def2-SVP DFT level.

Although estimating QC properties in any given conformation is important, it is even more important to estimate them in low-energy conformations (conformers), traditionally obtained with an iterative optimization process that utilizes a computational method at every optimization step. This process is called conformational (geometry) optimization or relaxation. Multiple calls to the computational method make geometry optimization extremely expensive in terms of computations, but it can be sped up by using an NNP instead [44–47]. GOLF paper [46] emphasizes that training such neural networks requires a large amount of data comprised of geometry optimization trajectories. To facilitate research on NN-based conformational optimization of druglike molecules, we extend our dataset with geometry optimization trajectories for 60,226 conformations of 16,974 molecules from the $\nabla^2$DFT. Together with these trajectories, the dataset contains 15,716,667 conformations in total.

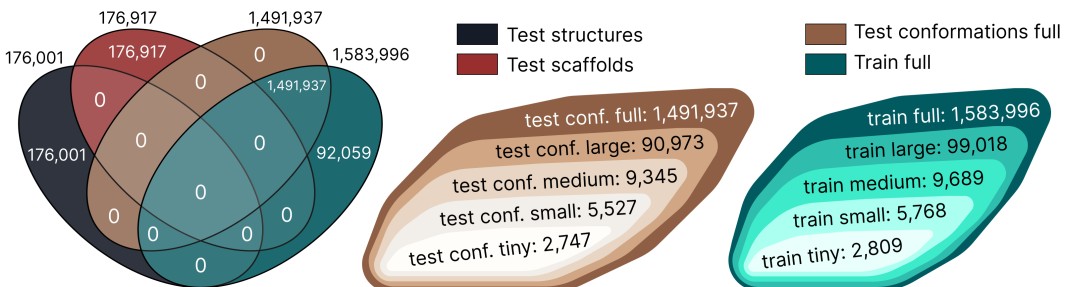

Figure 2: The figure illustrates the structure of $\nabla^2$DFT, which includes 12 predefined training and test splits designed for agile experimental design. Conformational test splits contain molecules that are also in the training splits, testing the models' ability to generalize to unseen molecular geometries. In contrast, Scaffold and Structure test sets are entirely independent of the training splits, evaluating the models' ability to generalize to completely new molecules.

A complete set of properties provided for each conformation, together with a unique dataset of relaxation trajectories for druglike molecules, make the $\nabla^2$DFT dataset *universal*.

In addition to the dataset, we propose a benchmark to evaluate the performance of NN-based models for QC and a framework that contains adaptations of 10 models, including the current state of the art. The framework is designed to be extendable and easy to use. The benchmark covers three important tasks in QC: Hamiltonian prediction (see Section 4.1), potential energy and atomic forces prediction (Section 4.2), and conformational optimization (Section 4.3). We implement Hamiltonian prediction models and NNPs within the proposed framework and carefully evaluate them on these three tasks. Models that directly predict wavefunctions, while promising, do not currently scale to large atomic systems and have generalization issues (see Section 2), so we leave their implementation and evaluation for future work. We highlight the **contributions** of this paper as follows: (1) a **universal dataset** that includes more than 30 QC properties such as energies, forces, Hamiltonians and overlap matrices, wavefunction objects, and optimization trajectories for druglike molecules; (2) a **comprehensive benchmark** for evaluating quantum chemistry models, encompassing tasks such as Hamiltonian prediction, energy and force prediction, and conformational optimization, with 12 predefined training and test splits for agile experimental design to assess how model performance depends on the available data and generalization to unseen geometries and novel molecules (see Figure 2); (3) an **extendable framework**[1] that contains adaptations of 10 quantum chemistry models together with reported benchmark metrics and checkpoints.

## 2 Related work

We begin with related datasets, grouped below according to the source of the original chemical information; detailed information is summarized in Table 1 and Table 9 in the Appendix.

**GDB-11/GDB-13/GDB-17.** QM7 [48], QM7b [49], QM8 [50], and QM9 [51] comprise one of the first families of datasets for ML research in computational chemistry. QM9 is the largest, with 130,000 small molecules in 5 atom types. However, QM9 includes only one low-energy conformation (conformer) per molecule, does not provide atomic forces, and only contains 5 atom types. QH9 [52] is a version of the QM9 dataset that provides Hamiltonians. MultiXC-QM9 [53] is a version of QM9 that provides energies calculated with several basis sets and exchange-correlation functionals. QM7-X [54] provides forces and contains several conformations per molecule, but it is limited to 7 heavy atoms and has ~7000 unique molecules. QM1B [55] contains larger molecules and provides 1 billion conformations but does not provide forces. The ANI-1 [56], ANI-1x, ANI-1ccx [57] family exceeds QM9 both in the number of conformations and size of molecules: ANI-1x has ~20M conformations for 57,000 molecules, and provides forces; but the ANI-XX family only has 4 atom types. GEOM [58] is a dataset of conformers for 450K molecules with ~37M conformations, but most computations were performed at a less accurate semi-empirical level of theory.

---

[1]$\nabla^2$DFT is available at https://github.com/AIRI-Institute/nablaDFT

Table 1: Summary of quantum chemistry datasets; ∗ – not provided directly but can be derived.

| | $\nabla^2$DFT (our) | $\nabla$DFT | QM7 | QM7b | QM7-X | QM9 | MultiXC-QM9 | QM1B | QH9 |
|---|---|---|---|---|---|---|---|---|---|
| Chem. inf. source | MOSES, ZINC21 (Zinc Clean Leads) | MOSES, ZINC21 (Zinc Clean Leads) | GDB-13 | GDB-13 | GDB-13 | GDB-17 | GDB-17 | GDB-11 | GDB-13, QM9 |
| # molecules | 2M | 1M | 7K | 7K | 7K | 134K | 134K | 1M | 130K |
| # conformers | 16M | 5M | 7K | 7K | 4M | 134K | 134K | 1B | 130K |
| # atoms | 8-62 | 8–62 | 1-23 | 1-23 | 1-23 | 3-29 | 3-29 | 9-11 | 3-29 |
| # heavy atoms | 8-27 | 8–27 | 1-7 | 1-7 | 1-7 | 1-9 | 1-9 | — | 1-9 |
| Atoms | H,C,N,O,S,Cl,F,Br | H,C,N,O,Cl,F,Br | H,C,N,O,S | H,C,N,O,S | H,C,N,O,S,Cl | H,C,N,O,F | H,C,N,O,F | H,C,N,O,F | H,C,N,O,F |
| Forces | ✓ | ✗* | ✗ | ✗ | ✓ | ✗ | ✗ | ✗ | ✗ |
| Hamiltonians | ✓ | ✓ | ✗ | ✗ | ✗ | ✗ | ✗ | ✗ | ✓ |
| Optim. traj. | ✓ | ✗ | ✗ | ✗ | ✗ | ✗ | ✗ | ✗ | ✗ |
| Basis set and XC-func. | ΩB97X- D/Def2- SVP | ΩB97X- D/Def2- SVP | PBE0 | ZINDO, SCS, PBE0, GW | ePBE0+MBD, TotFOR | B3LYP/6-31G(2df,p)+ G4MP2 | B3LYP,PBE/6–31G(2df,p), SZ,DZP,TZP | STO-3G/B3LYP | B3LYP/Def2SVP |
| Storage size | 220 Tb | 100 Tb | 17.9 MB | 16.1 MB | 1.35 Gb | 230 Mb | 317 Gb | 240 GB | 28.4 Gb |
| | GEOM | ANI-1 | ANI-1x/ANI-1cxx | OrbNet Denali | QMugs | SPICE | PubChemQC | Frag20 | VQM24 |
| Chem. inf. source | QM9, AICures | GDB | GDB-11, ChEMBL | ChEM-BL27 | ChEMBL | PubChem+, DES370K+, other | PubChem | PubChem, ZINC | Combinatorial |
| # molecules | 450K | 57K | 57K | 16K | 665K | 19K | 85M | 565K | 10K |
| # conformers | 37M | 24M | 20M | 2.3M | 2M | 1KK | 85M | 566K | 835K |
| # atoms | — | — | 2-26 | — | 4-228 | 2–96 | — | — | 4,5 |
| # heavy atoms | — | 1-11 | 1-8 | — | 4-100 | — | 51 | 20 | — |
| Atoms | H,C,N,O,F | H,C,N,O | H,C,N,O | H,Li,B,C,N,O,F,Na,Mg,Si,P,S,Cl,K,Ca,Br,I | H,C,N,O,P,S,Cl,F,Br,I | H,Li,C,N,O,F,Na,Mg,P,S,Cl,K,Ca,Br,I | H,C,N, O,P,S,F,Cl,Na,K,Mg,Ca | H,B,C,O,N,F,P,S,Cl,Br | H,C,N,O,F,Si,P,S,Cl,Br |
| Forces | ✗ | ✗ | ✓ | ✗ | ✗* | ✓ | ✗ | ✗ | ✗ |
| Hamiltonians | ✗ | ✗ | ✗ | ✗ | ✗* | ✗ | ✗ | ✗ | ✗ |
| Optim traj. | ✗ | ✗ | ✗ | ✗ | ✗ | ✗ | ✗ | ✗ | ✗ |
| Basis set and XC-func. | mTZVPP/R2scan-3 | WB97x/6–31g(d) | wB97x/6-31G*, wB97x/def2-TZVPP, CCSD(T)*/CBS | ΩB97X-D3/Def2-TZVP | ωB97X- D/Def2- SVP+GFN2-XTB | ωB97M-D3(BJ)/Def2-TZVPPD | B3LYP/6-31G*+PM6 | B3LYP/6-31G* | ΩB97X-D3/Cc- pVDZ, DMC@PBE0(cEC/Cc-VQZ)) |
| Storage size | 130.8 GB | 4.48 Gb | 5.29 Gb | 2.74 GB | 7 Tb | 37.5 GB | 100 Tb | 566 Mb | 1.5 Gb |

**ChEMBL/ChEMBL27**. OrbNet Denali [59] contains ~16,000 molecules, ~2,300,000 conformations, and 17 atom types; it does not provide forces and has a low molecule diversity. QMugs [60] includes ~665,000 molecules and ~2,000,000 conformers, 10 atom types, and up to 228 atoms per molecule. QMugs provides about 50 molecular properties and the density matrix but does not provide forces, and an additional step of the SCF solver is required to obtain the Hamiltonian.

**PubChem**. The latest version of PubChemQC [61] includes 85,938,443 molecules and a single conformer for each molecule, with up to 51 atoms per molecule and 13 atom types. This dataset does not provide forces and Hamiltonians but includes full solver convergence reports that can be parsed to obtain QC properties. The SPICE [62] dataset combines a subset of PubChem, DES370K, and some other sources of chemical information; the small molecule part of SPICE includes ~14,600 molecules, 730,000 conformations, 10 atom types, and provides information about forces. The main drawbacks include a small number of molecules and the lack of Hamiltonians.

**Combinatorial**. Instead of sampling molecules from a specific database, VQM24 [63] covers the full chemical space of molecules with up to 5 heavy atoms (~258,000 molecules and ~577,000 conformations) via a combinatorial algorithm with filtering. Additionally, conformers of 10,793 molecules with up to 4 heavy atoms are evaluated with quantum Monte Carlo.

**Other domains**. Other QC datasets include materials, chemical reactions, peptides, nanotubes, and more, or provide molecular dynamic (MD) trajectories. ISO17 [31] and MD17 [64] provide MD trajectories for several organic molecules. MD22 [39] is a dataset of MD trajectories for several large atomic structures. OC20 [47], OC22 [44], and OC20-Dense [45] provide optimization trajectories for various adsorbate-catalyst pairs. PCQM4Mv2, a part of the Open Graph Benchmark project [65], contains a subset of properties from the PubChemQC dataset. QMOF [66] is a dataset of metal-organic substances. GeckoQ [67] is a dataset with atomic structures of atmospherically relevant

molecules. DES370K [68] is a dataset of dimers computed with the CCSD(T) level of theory. Transition1x [69] is a dataset of molecules and reaction pathways.

The key feature of $\nabla^2$DFT that separates it from other datasets is its universality. It provides all the above-mentioned molecular properties, full `Psi4` wavefunction objects, Hamiltonian matrices, and geometry optimization trajectories, all calculated at a reasonably accurate DFT level for a large number of diverse molecules with 8 atom types and up to 62 atoms. Moreover, our source of chemical information is the MOSES dataset of structures of commercially available drug-like molecules [70] (ZINC Clean Leads [71]), which makes $\nabla^2$DFT most relevant for the chemical and pharmaceutical industry.

**Neural Network Potentials**. NNPs are a family of models that predict potential energies and atomic forces based on conformations; NNPs are useful for downstream applications and represent the primary focus of our benchmark. Most existing models [31–40] are based on message passing in NNs [16]. Importantly, NNPs can perform molecular dynamics (MD) by predicting forces.

**Wavefunction learning and Hamiltonian prediction**. *PauliNet* [17] and *FermiNet* [18] are two deep learning wavefunction Ansätze that provide nearly exact solutions to the electronic Schrödinger equation for single atoms and small molecules such as LiH, ethanol, and bicyclobutane. However, scaling these approaches to larger systems and increasing the number of molecules presents significant challenges: training *PauliNet* and *Ferminet* requires a substantial amount of computation even for a system of two nitrogen atoms [19]. Recently proposed models [19–24, 72, 73] employ Transformers, adapter models structure, and other approaches to get better accuracy and generalization, but still can deal with only small structures and need lots of compute; nevertheless, the $\nabla^2$DFT dataset provides all necessary data for the training of such models. Instead of directly learning a wavefunction, one can predict Hartree-Fock or DFT Hamiltonian matrices. *SchNOrb* [25] is a direct continuation of the *SchNet* model [31]. *PhiSNet* [26] can be seen as a SE(3)-Equivariant variation of *SchNOrb*, which makes it more accurate and stable. *DeepH* [27] and *DeepH-E3* [28] are similar models which can be applied to crystal structures. Finally, the works [29, 30] propose variations of *PhiSNet* tested not only in a single molecule scenario but also on *QM9*-based datasets. Considering this a promising approach, we add three of these models to our benchmark to test their performance and generalizability in a more difficult scenario.

**Geometry optimization with neural networks**. Guan et al. [74] and Lu et al. [75] frame the conformation optimization problem as a conditional generation task, training models to generate low-energy conformations conditioned on conformations generated by RDKit or randomly sampled from pseudo-optimization trajectories by minimizing RMSD between predicted and real atom coordinates. Another approach [44–46] is to use interatomic forces, predicted by an NNP, as antigradients for an optimization method such as L-BFGS [76]. Tsypin et al. [46] investigate the iterative optimization of molecules and demonstrate that NNPs can match the optimization quality of DFT-based methods by utilizing extensive datasets comprised of geometry optimization trajectories. In this work, we augment $\nabla^2$DFT with geometry optimization trajectories and establish a new benchmark to support further research on the iterative optimization of molecular conformations with NNPs.

## 3 Dataset

The primary contribution of this work is $\nabla^2$**DFT**, an extension of the large-scale **nablaDFT** dataset of QC properties for druglike molecules [11]. $\nabla^2$**DFT** is based on the Molecular Sets (MOSES) dataset [70]; it contains 1,936,929 molecules with atoms C, N, S, O, F, Cl, Br, and H, 448,854 unique Bemis-Murcko scaffolds [77], and 58,315 unique BRICS fragments [78].

For each molecule from the dataset, we have run the conformation generation method from the *RDKit* software suite [79] proposed by Wang et al. [80], getting 1 to 100 conformations per molecule. Next, we clustered the resulting conformations with the Butina clustering method [81], selected the minimal set of clusters that cover 95% of the conformations, and included their centroids as conformations in $\nabla^2$**DFT**, obtaining 1 to 69 unique conformations for each molecule, with 12,676,264 total conformations in the full dataset. For each conformation, we calculated its electronic properties, including the total energy (E), interatomic forces $F$, DFT Hamiltonian matrix (H), and DFT overlap matrix (S) (see the full list in Table 2). All properties were calculated using the Kohn-Sham method [82] at $\omega$B97X-D/def2-SVP level of theory using the quantum-chemical software *Psi4* [83], version 1.5, with default parameters: the Lebedev-Treutler grid with a Treutler partition of the atomic

Table 2: Properties available for each data instance in $\nabla^2$DFT.

| Access mode | Content |
|---|---|
| Main data (basic dataloader) | Atom numbers, atom positions, energy ('DFT FORMATION ENERGY'), forces, Hamiltonian (Fock matrix), overlap matrix, coefficients matrix |
| Metainformation | 'DFT TOTAL ENERGY', 'DFT XC ENERGY', 'DFT NUCLEAR REPULSION ENERGY', 'DFT ONE-ELECTRON ENERGY', 'DFT TWO-ELECTRON ENERGY', 'DFT DIPOLE X', 'DFT DIPOLE Y', 'DFT DIPOLE Z', 'DFT TOTAL DIPOLE', 'DFT ROT CONSTANT A', 'DFT ROT CONSTANT B', 'DFT ROT CONSTANT C', 'DFT HOMO', 'DFT LUMO', 'DFT HOMO-LUMO GAP', 'DFT ATOMIC ENERGY', 'DFT FORMATION ENERGY' |
| Raw stored wavefunction object | *All data from two previous rows.* Ca/Cb (molecular orbital coefficients), Da/Db (density matrix), Fa/Fb (Fock matrix), H (Core Hamiltonian), S (overlap matrix), X (XC-functional matrix), aotoso (Atomic Orbital to Symmetry Orbital), epsilon_a/epsilon_b (orbital eigenvalues), SCF DIPOLE, doccpi (number of doubly occupied orbitals), nmo (number of molecule orbitals), 'DISPERSION CORRECTION ENERGY', 'GRID ELECTRONS TOTAL' |
| Available after loading into Psi4 | *All data from three previous rows.* Electric dipole moment, Electric quadrupole moment, All moments up order N, Electrostatic potential at nuclei, Electrostatic potential on grid, Electric field on grid, Molecular orbital extents, Mulliken atomic charges, Löwdin atomic charges, Wiberg bond indices, Mayer bond indices, Natural orbital occupations, Stockholder Atomic Multipoles, Hirshfeld volume ratios |

weights, 75 radial points and 302 spherical points, convergence of energy and density up to $10^{-6}$ as the criterion for SCF cycle termination, and $10^{-12}$ as the integral calculation threshold.

We applied a multi-step filtration protocol to ensure the validity of the provided QC computations. We filtered 31 molecules from the MOSES dataset where the above procedure could not produce valid conformations. Then, we filtered samples with anomalous values of QC properties: *('DTF TOTAL ENERGY' > 0), ('DFT TOTAL DIPOLE' < 20),* or *('DFT FORMATION ENERGY' < 0)*. We excluded 29 conformations and discarded 2 more molecules, totaling 33. Finally, 17 additional molecules and 145 conformations with an atomic forces norm exceeding 99.999 percentile were discarded.

To set up the $\nabla^2$DFT benchmark, we provide several data splits that can be used to compare different models fairly (see Fig. 2). First, we fix the training set $\mathcal{D}^{\text{full}}$ that consists of 1,583,996 molecules with 8,849,983 conformations and its smaller subsets $\mathcal{D}^{\text{large}}$, $\mathcal{D}^{\text{medium}}$, $\mathcal{D}^{\text{small}}$, and $\mathcal{D}^{\text{tiny}}$ with 99,018, 9689, 5768, and 2809 molecules and 500,552, 49,725, 28,362, and 12,145 conformations respectively. These subsets help study how the performance of various models depends on available data.

We select 176,001 random molecules, not present in $\mathcal{D}^{\text{full}}$, and call it the *structure test set* $\mathcal{D}^{\text{structure}}$. We also select another 176,917 molecules containing a Bemis-Murcko scaffold, which are not present in $\mathcal{D}^{\text{full}}$, and call it the *scaffold test set* $\mathcal{D}^{\text{scaffold}}$. Finally, for each training set we have the corresponding *conformation test set* that contains different conformations of the same molecules: $\mathcal{D}^{\text{conf-full}}$, $\mathcal{D}^{\text{conf-large}}$, $\mathcal{D}^{\text{conf-medium}}$, $\mathcal{D}^{\text{conf-small}}$, and $\mathcal{D}^{\text{conf-tiny}}$, with 1,491,937, 90,973, 9345, 5527, 2747) molecules and 1,542,971, 93,530, 9532, 5634, 2774 conformations respectively (the sizes are different from training sets because molecules with a single conformation cannot appear here). Conformation test sets are designed to test the ability of the models to generalize to unseen *geometries* of molecules; structure and scaffold test sets, to unseen *molecules*. We expect the conformation test set to be the easiest and the scaffold test set to be the most challenging as it contains unseen molecular fragments.

We also present the second dataset based on $\nabla^2$DFT: $\nabla^2$DFT$_{\text{opt}}$ that contains relaxation trajectories for approximately 60K conformations of 17K molecules, resulting in approximately 3M geometries. We report the energy (E) and forces matrix (F) for each geometry from these trajectories. We split the trajectories data into 3 datasets: $\mathcal{D}^{\text{traj-test}}$, $\mathcal{D}^{\text{traj-medium}}$, and $\mathcal{D}^{\text{traj-additional}}$: $\mathcal{D}^{\text{traj-test}}$ is designed for fast validation of NNPs for conformation optimization contains optimization trajectories of 1000 molecules from $\mathcal{D}^{\text{structure}}$ and 1000 molecules from $\mathcal{D}^{\text{scaffold}}$, $\mathcal{D}^{\text{traj-medium}}$ contains trajectories for 9538 molecules from $\mathcal{D}^{\text{medium}}$, and $\mathcal{D}^{\text{traj-additional}}$ provides additional optimization trajectories for 5462 molecules, suitable both for training and validation. As part of the benchmark, we provide databases

Table 3: Prediction metrics for Hamiltonian and overlap matrices; mean absolute error, less is better.

| | Model | Hamiltonian prediction MAE, $\times 10^{-3} E_h$ | | | | Overlap prediction MAE, $\times 10^{-5}$ | | | |
| --- | --- | --- | --- | --- | --- | --- | --- | --- | --- |
| | | $\mathcal{D}^{\text{tiny}}$ | $\mathcal{D}^{\text{small}}$ | $\mathcal{D}^{\text{medium}}$ | $\mathcal{D}^{\text{large}}$ | $\mathcal{D}^{\text{tiny}}$ | $\mathcal{D}^{\text{small}}$ | $\mathcal{D}^{\text{medium}}$ | $\mathcal{D}^{\text{large}}$ |
| **Structure test split** | SchNOrb | 19.8 | 19.6 | 19.6 | 19.8 | 1320 | 1310 | 1320 | 1340 |
| | PhiSNet | **0.19** | **0.32** | **0.34** | **0.36** | **2.7** | **3.0** | **2.9** | **3.3** |
| | QHNet | 0.98 | 0.79 | 0.52 | 0.69 | - | - | - | - |
| **Scaffolds test split** | SchNOrb | 19.9 | 19.8 | 20. | 19.9 | 1330 | 1320 | 1330 | 1340 |
| | PhiSNet | **0.19** | **0.32** | **0.34** | **0.36** | **2.6** | **2.9** | **2.9** | **3.2** |
| | QHNet | 0.98 | 0.79 | 0.52 | 0.69 | - | - | - | - |
| **Confor- mations test split** | SchNOrb | 21.5 | 20.7 | 20.7 | 20.6 | 1410 | 1360 | 1370 | 1370 |
| | PhiSNet | **0.18** | **0.33** | **0.35** | **0.37** | **3.0** | **3.2** | **3.1** | **3.5** |
| | QHNet | 0.84 | 0.73 | 0.52 | 0.68 | - | - | - | - |

for each subset and task and a complete archive with wavefunction files produced by *Psi4* that contain QC properties of the corresponding molecule and can be used in further computations.

# 4 Benchmark setup and results

The goal of our benchmark is to advance and standardize studies in the field of machine learning methods for computational quantum chemistry. We focus on three fundamental tasks: (1) DFT Hamiltonian matrix prediction, (2) molecular conformation energy and atomic forces prediction, and (3) conformational optimization. In the first two tasks, we measure the ability of state-of-the-art models to generalize across a diverse set of molecules. In the third, we evaluate the performance of NNPs trained to predict energies and atomic forces for conformational optimization. For the Hamiltonian prediction task we compare *SchNOrb* [25], *PhiSNet* [26] and *QHNet* [29] models; note that we predict full Hamiltonian (Fock) matrices, while Khrabrov et al. [11] predicted core Hamiltonian matrices. For the energy and atomic forces prediction, we compare linear regression, *SchNet* [31], *SchNOrb*, *Dimenet++* [36], *PaiNN* [34], *Graphormer3D* [84], *GemNet-OC* [37, 38], *EquiformerV2* [40], and *eSCN* [85]. All models have been trained on $\mathcal{D}^{\text{tiny}}, \mathcal{D}^{\text{small}}, \mathcal{D}^{\text{medium}}, \mathcal{D}^{\text{large}}$ subsets of $\nabla^2$DFT and evaluated on $\mathcal{D}^{\text{structure}}, \mathcal{D}^{\text{scaffold}}, \mathcal{D}^{\text{conformation}}$ with mean absolute error (MAE), as shown in (2), (3), (4). Details of the training procedure are given in Appendix A.

## 4.1 Hamiltonian matrix prediction

Neural networks are trained to minimize $L_1$ or $L_2$ loss for the Hamiltonian matrix $\mathbf{H} \in \mathbb{R}^{n_s \times n_s}$, where $n_s$ is the number of electronic orbitals for a conformation $s$. PhiSNet and SchNOrb are also trained to predict the overlap matrix $S$, and SchNOrb predicts energy. A comparison of the MAE metrics (see Appendix B.3) is reported in Table 3. *PhiSNet* performs best, even though training did not converge for larger splits (we stopped training after 1920 GPU hours). The *SchNOrb* model benefits vastly from the additional energy prediction task, compared with [11], but still performs worse than *PhiSNet* and *QHNet*, which agrees with the results of a single molecule setup [26].

The models perform better on the conformation test splits $\mathcal{D}^{\text{conf}}$; this is expected because the training set is more similar to the test set in this case. Hamiltonian prediction results on $\nabla^2$DFT are worse than previously published; e.g., *PhiSNet* has MAE $1.8 \times 10^{-5} E_h$ for the molecules from *MD17* [26]; *QHNet*, $7 \times 10^{-5} E_h$ on QH9 [52]. We believe this is caused by a higher diversity of $\nabla^2$DFT, and our benchmark highlights generalization issues of Hamiltonian prediction models.

## 4.2 Energy and atomic forces prediction

We denote the DFT energy for conformation $s$ as $E_s$. For energy and atomic forces prediction, a neural network takes a conformation $s$ as input and outputs the energy $\hat{E}_s = f(s; \boldsymbol{\theta})$, $f(s; \boldsymbol{\theta}) : \{\boldsymbol{z}, \boldsymbol{X}\} \to \mathbb{R}$. To predict interatomic forces, we take the gradients of $\hat{E}_s$ w.r.t. coordinates of atoms $\boldsymbol{X}$ in case of *SchNet*, *PaiNN* and *DimeNet++*: $\hat{\boldsymbol{F}}_s = \frac{\partial f(s; \boldsymbol{\theta})}{\partial \boldsymbol{X}}$, $\hat{\boldsymbol{F}}_s \in \mathbb{R}^{n \times 3}$, where $n$ is the number of atoms in the system. For *EquiformerV2*, *Graphormer3D*, *GemNet-OC* and *eSCN* we use a separate head:

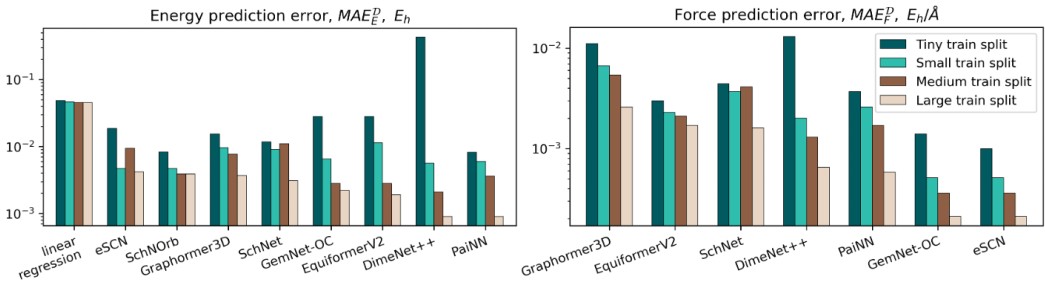

Figure 3: Performance of neural networks on $\mathcal{D}^{\text{structures}}$ test split. Colours of the bars show the training splits ($\mathcal{D}^{\text{tiny}}$, $\mathcal{D}^{\text{small}}$, $\mathcal{D}^{\text{medium}}$, $\mathcal{D}^{\text{large}}$). Y-axis is log-scales.

$\hat{\boldsymbol{F}}_s = F(s;\boldsymbol{\phi}), F(s;\boldsymbol{\phi}) : \{\boldsymbol{z}, \boldsymbol{X}\} \to \mathbb{R}^{n\times3}$. NNs are trained to minimize $L_1$, $L_2$, or RMSE loss for energy and interatomic forces. We compare the models based on MAE (see Appendix B.3).

Metrics for energy and forces prediction are reported in Figure 3 and Table 7, respectively. First, a significant improvement of all metrics compared to [11] is because we subtracted atomization energies from the target energies (see Appendix B.4 for details). Together with a richer dataset, this has led to a drastic decrease in energy and forces prediction error. Second, we see that all models benefit from increasing the size of the training dataset, which highlights the need to develop extensive datasets for NNPs. Third, both Transformer-based models perform worse than MPNN-based ones, indicating that additional architecture search and hyperparameter tuning are likely needed for such models. Moreover, we note that models with a separate force prediction head work better in the forces prediction task but worse for energy prediction. We hypothesize that this discrepancy is caused either by the gradient of the forces prediction loss serving as a regularizer or by the improper choice of hyperparameters. Finally, we see that SchNOrb shows performance on par with state-of-the-art networks for energy prediction, which supports the conjecture that modeling the electronic structure helps NN models in property prediction tasks. In our setup, NNPs perform worse but comparable with previously known benchmarks such as *QM9* or *MD17*; e.g., DimeNet++ has MAE $23 \times 10^{-5} E_h$ on *QM9* [36]. Decreased performance is likely caused by more diverse and larger molecules in $\nabla^2$DFT and limited sizes of training sets. We leave training NNPs on the full dataset for future work.

### 4.3 Conformation optimization

To evaluate the quality of optimization with NNPs, we use a fixed subset $\mathcal{D}^{\text{test-traj}}$ of $\nabla^2$DFT that shares no molecules with any train sets. For each conformation $s \in \mathcal{D}^{\text{test-traj}}$ we perform optimization with the DFT simulator to get the ground truth optimal conformation $s_{\textbf{opt}}$ and its energy $E_{s_{\textbf{opt}}}$. For the conformation optimization task we use the metrics $\overline{\text{pct}}_T$, $\text{pct}_{\text{success}}$ and $\text{pct}_{\text{div}}$ proposed in [46] (see Appendix B.3). We benchmark NNPs and additionally two non-neural methods: RDKit's MMFF and a semi-empirical xTB, chosen as a baseline because it is often used for conformational optimization in other datasets [58, 60].

Geometry optimization results are shown in Tables 4, 5. First, the optimization quality of NNPs drastically increases with the increase in the training dataset size. Second, the quality of forces prediction directly influences the quality of conformational optimization (see Fig. 3 and Table 4). We conclude that to obtain the best architecture for optimization, one can choose the best-performing model based on the results of forces prediction.

While optimization performance is relatively good for several models trained on the $\mathcal{D}^{large}$, it can be further improved by incorporating optimization trajectories into training [46]. We show this by finetuning our best-performing optimization models (*PaiNN* and *GemNet-OC*) on $\mathcal{D}^{\text{medium}}$. We call this models *PaiNN*-finetune and *GemNet-OC*-finetune and additionally compare them with non-neural methods (see Table 5). We have also calculated the RMSD between final conformations and ground-truth optimized conformations $s_{\text{opt}}$. The RMSD distribution is shown in Fig. 4 (see also Appendix D). We observe that NNPs significantly outperform non-neural methods in terms of $\text{pct}_{\text{success}}$ while being comparable in terms of RMSD. This result confirms that the widely used RMSD is not an ideal metric for geometry optimization.

Table 4: Geometry optimization metrics for NNPs.

| Metric | Model | $\mathcal{D}^{\text{tiny}}$ | $\mathcal{D}^{\text{small}}$ | $\mathcal{D}^{\text{medium}}$ | $\mathcal{D}^{\text{large}}$ |
|---|---|---|---|---|---|
| $\overline{\text{pct}}_T$ (%) ↑ | SchNet | 38.56 | 39.75 | 36.50 | 75.51 |
| | PaiNN | 60.26 | 66.63 | 74.16 | 98.50 |
| | DimeNet++ | 32.27 | **89.16** | **93.22** | 96.35 |
| | EquiformerV2 | 64.41 | 76.11 | 75.24 | 86.10 |
| | eSCN | **76.83** | 85.94 | 89.34 | 97.27 |
| | GemNet-OC | 69.04 | 85.57 | 92.42 | **100.06** |
| $\overline{\text{pct}}_{\text{success}}$ (%) ↑ | SchNet | 0.0 | 0.0 | 0.0 | 4.00 |
| | PaiNN | 0.0 | 0.11 | 2.6 | 77.09 |
| | DimeNet++ | 0.0 | 13.02 | **34.04** | 55.71 |
| | EquiformerV2 | 6.90 | 12.62 | 16.38 | 32.01 |
| | eSCN | **11.49** | **19.23** | 25.39 | 53.38 |
| | GemNet-OC | 0.91 | 10.42 | 30.94 | **90.71** |
| $\overline{\text{pct}}_{\text{div}}$ (%) ↓ | SchNet | 39.6 | 34.85 | 45.82 | 0.8 |
| | PaiNN | 21.25 | 10.35 | 7.00 | **0.05** |
| | DimeNet++ | 96.55 | 20.50 | 7.6 | 1.00 |
| | EquiformerV2 | 92.75 | 84.55 | 84.75 | 76.10 |
| | eSCN | 59.1 | 27.7 | 11.00 | 0.80 |
| | GemNet-OC | **11.55** | **0.75** | **0.60** | 0.40 |

Table 5: Geometry optimization metrics for *PaiNN*, *GemNet-OC*, and computational approaches

| Metrics | PaiNN | PaiNN-finetune | GemNet-OC | GemNet-OC-finetune | RDKit MMFF | xTB |
|---|---|---|---|---|---|---|
| $\overline{\text{pct}}_T$(%) ↑ | 98.50 | 99.83 | **100.06** | 100.01 | 84.44 | 92.33 |
| $\text{pct}_{\text{success}}$(%) ↑ | 77.09 | 84.35 | 90.71 | **94.55** | 1.9 | 3.1 |
| $\text{pct}_{\text{div}}$(%) ↓ | 0.05 | **0.** | 0.4 | **0.** | **0.** | **0.** |
| $\overline{\text{RMSD}}$ ↓ | $.50 \pm .53$ | $.52 \pm .54$ | $.73 \pm .54$ | $.38 \pm .54$ | $.72 \pm .54$ | $.52 \pm .51$ |

## 5   Limitations

$\nabla^2$DFT does not contain solvated molecules or protein-ligand pairs (important for ML applications in drug design). It lacks charged and open-shell systems, nano-particles, nanotubes, big rings, and other non-drug-like structures. Moreover, $\nabla^2$DFT is unsuitable for material science and inorganic chemistry and for ML-based studies of long-range and non-covalent interactions.

## 6   Conclusion

This work introduces $\nabla^2$DFT, a universal dataset of drug-like molecules for quantum chemistry models. It contains ~16 million conformations of ~2 million molecules, with key properties such as energy, forces, and Hamiltonian matrices. A unique property of $\nabla^2$DFT is relaxation trajectories for ~60,000 conformations of ~17,000 molecules, aiding conformational optimization research. We propose a novel benchmark for evaluating quantum chemistry models and an extendable framework for training them. Our experiments highlight the importance of training on large datasets and emphasize the need for further dataset development.

## Acknowledgments

This work was supported by a grant for research centers in the field of artificial intelligence, provided by the Analytical Center for the Government of the Russian Federation in accordance with the subsidy agreement (agreement identifier 000000D730321P5Q0002) and the agreement with the Ivannikov

Institute for System Programming of the Russian Academy of Sciences dated November 2, 2021 No. 70-2021-00142.

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

Table 6: Parameters number and compute for GNN models

| Backbone | Parameters number | GPU hours for training | FLOPS |
|---|---|---|---|
| **Neural Network Potentials** | | | |
| *SchNet* | 0.5M | 504 | 1.9e19 |
| *PaiNN* | 1.3M | 720 | 2.5e19 |
| *DimeNet++* | 5.1M | 600 | 2.8e19 |
| *Graphormer3D* | 10.7M | 605 | 1.7e19 |
| *GemNet-OC* | 37.8M | 3046 | 1.08e20 |
| *EquiformerV2* | 83.1 M | 2016 | 7.0e19 |
| *eSCN* | 34.3 M | 2016 | 9.3e19 |
| **Hamiltonian Prediction Models** | | | |
| *SchNorb* | 242.36M | 1920 | 9.1e19 |
| *PhiSNet* | 21M | 1920 | 6.9e19 |
| *QHNet* | 21.9M | 4378 | 7.8e21 |

# A  Computational and experimental setup

All DFT computations were carried out with Psi4 software on Intel(R) Xeon(R) Gold 2.60Hz CPU-cores, and the total computational cost is $\approx$ 120 CPU-years. All GPU computations were carried out on NVIDIA V100/A100 graphical units.

In general, model hyperparameters were derived from the corresponding publications and can be found in https://github.com/AIRI-Institute/nablaDFT/tree/main/config/model. We list an approximate compute needed for training models on $\mathcal{D}^{large}$ in the Table 6. *SchNet* and *PaiNN* models code was based on Schnetpack2.0 [86], plus we provide a PyTorch Geometric [87] version of *PaiNN*, based on the Open Catalyst [47] codebase. For *DimeNet++* we used an implementation from PyTorch Geometric. For *SchNOrb*, *PhiSNet* and *Graphormer3D* we used an adaptation of the code, provided by the papers authors. Finally, for *GemNet-OC*, *EquformerV2* and *eSCN* models we adapted the code from the Open Catalyst [47] codebase.

# B  Preliminaries

## B.1  Conformations

Conformations represent structural arrangements of the same molecule, distinguished by rotations around single bonds and bond stretching. A conformation $s = \{z, X\}$ of a molecule is defined by a set of atomic numbers $z = \{z_1, \ldots, z_n\}, z_i \in \mathbb{N}$ and atomic coordinates $X = \{x_1, \ldots, x_n\}, x_i \in \mathbb{R}^3$, where $n$ denotes the number of atoms in the molecule. Given that most druglike molecules are capable of adopting multiple conformations, conformational analysis becomes a pivotal component in molecular modeling. This is because a molecule's biological activity and physico-chemical properties are largely determined by its specific conformation. Conformational analysis entails the exploration of the total energies of various conformations for a particular molecule (conformational energies $E_s$).

## B.2  DFT

Anti-symmetrized products of single-electron functions or molecular orbitals are frequently used in quantum chemistry to express the electronic wavefunction $\Psi$ associated with the electronic time-independent Schrödinger equation $\hat{H}\Psi = E\Psi$.

These single-particle functions are usually defined in a local atomic orbital basis of spherical atomic functions $|\psi_m\rangle = \sum_i c_m^i |\phi_i\rangle$, where $|\phi_i\rangle$ are the basis functions and $c_m^i$ are the coefficients. As a result, one can represent the electronic Schrödinger equation in matrix form as

$$\mathbf{F}_\sigma \mathbf{c}_\sigma = \epsilon_\sigma \mathbf{S}\mathbf{c}_\sigma,$$

where $\mathbf{F}$ is the Fock matrix (otherwise called the Hamiltonian matrix $\mathbf{H}$), $\mathbf{H}_{ij} = \left\langle \phi_i \mid \hat{H} \mid \phi_j \right\rangle$, $\mathbf{S}$ is the overlap matrix, $\mathbf{S}_{ij} = \langle \phi_i \mid \phi_j \rangle$, $\mathbf{c}$ is the vector of coefficients, and $\sigma = \{\alpha, \beta\}$ is the spin index.

In matrix form, the single-particle wavefunction expansion can be represented by using Einstein summation as

$$\psi_i^\sigma(\vec{r}) = C_{\mu i}^\sigma \phi_\mu(\vec{r}).$$

Therefore, the density matrix is represented as

$$D_{ij}^\sigma = C_{ik}^\sigma C_{jk}^\sigma$$

In DFT, the matrix $\mathbf{F}$ corresponds to the Kohn-Sham matrix:

$$F_{ij}^\sigma = \mathrm{Hc}_{ij}^\sigma + J_{ij}^\sigma + V_{ij}^{\mathrm{xc}},$$

where $\mathrm{Hc}_{ij}^\sigma$ is the core Hamiltonian matrix, $J_{ij}^\sigma$ is the Coulomb matrix, and $V_{ij}^{\mathrm{xc}}$ is the exchange-correlation potential matrix.

In DFT, the total energy of the system (e.g., total energy of a conformation) can be expressed as

$$E_{\mathrm{total}} = D_{ij}^{\mathrm{T}}(T_{ij} + V_{ij}) + \frac{1}{2} D_{ij}^{\mathrm{T}} D_{\lambda\beta}^{\mathrm{T}}(ij|\lambda\beta) + E_{\mathrm{xc}}[\rho_\alpha, \rho_\beta],$$

where $T$ is the noninteracting quasiparticle kinetic energy operator, $V$ is the nucleus-electron attraction potential, $D$ is the total electron density matrix, and $E_{\mathrm{xc}}$ is the (potentially nonlocal) exchange, correlation, and residual kinetic energy functional. The residual kinetic energy term is usually quite small and is often incorporated in the correlation term of $E_{\mathrm{xc}}$.

One can represent the Hamiltonian matrix in block form [25]:

$$\mathbf{H} = \begin{bmatrix} \mathbf{H}_{11} & \cdots & \mathbf{H}_{1j} & \cdots & \mathbf{H}_{1n} \\ \vdots & \ddots & \vdots & & \vdots \\ \mathbf{H}_{i1} & \cdots & \mathbf{H}_{ij} & \cdots & \mathbf{H}_{in} \\ \vdots & & \vdots & \ddots & \vdots \\ \mathbf{H}_{n1} & \cdots & \mathbf{H}_{nj} & \cdots & \mathbf{H}_{nn} \end{bmatrix}$$

Here the matrix block $\mathbf{H}_{ij} \in \mathbb{R}^{n_{\mathrm{ao},i} \times n_{\mathrm{ao},j}}$ and the choice of $n_{\mathrm{ao},i}$ and $n_{\mathrm{ao},j}$ atomic orbitals depend on the atoms $i,j$ within their chemical environments. This fact underlies the construction of interaction modules in the NN Hamiltonian prediction models: they construct representations of atom pairs from representations of atomic environments.

Unfortunately, eigenvalues and wavefunction coefficients are not well-behaved or smooth functions because they depend on atomic coordinates and changing molecular configurations. This problem can be addressed by deep learning architectures that directly define the Hamiltonian matrix.

We define the interatomic forces $F_s \in \mathbb{R}^{n \times 3}$ as the gradient of $E_s^{\mathrm{total}}$ for conformation $s$ w.r.t. the Euclidian coordinates $\boldsymbol{X}$:

$$F_s = \frac{\partial E_s^{\mathrm{total}}}{\partial \boldsymbol{X}}. \tag{1}$$

### B.3   Metrics

We use the following metrics for model validation:

$$\mathrm{MAE}_E^{\mathcal{D}} = \frac{1}{|\mathcal{D}|} \sum_{s \in \mathcal{D}} |\hat{E}_s - E_s|, \tag{2}$$

$$\mathrm{MAE}_F^{\mathcal{D}} = \frac{1}{|\mathcal{D}|} \sum_{s \in \mathcal{D}} \|\hat{\boldsymbol{F}}_s - \boldsymbol{F}_s\|_1. \tag{3}$$

$$\mathrm{MAE}_H^{\mathcal{D}} = \frac{1}{|\mathcal{D}|} \sum_{s \in \mathcal{D}} \frac{1}{n_s^2} \sum_{i,j} |\hat{\mathbf{H}}_s^{ij} - \mathbf{H}_s^{ij}|. \tag{4}$$

The quality of the NNP-optimization is evaluated with the average percentage of minimized energy for terminal conformations $s_T$:

$$\overline{\mathrm{pct}}_T = \frac{1}{|\mathcal{D}^{\mathrm{test\text{-}traj}}|} \sum_{s \in \mathcal{D}^{\mathrm{test\text{-}traj}}} \mathrm{pct}(s_T) = \frac{1}{|\mathcal{D}^{\mathrm{test\text{-}traj}}|} \sum_{s \in \mathcal{D}^{\mathrm{test\text{-}traj}}} 100\% * \frac{E_{s_0} - E_{s_T}}{E_{s_0} - E_{s_{\mathrm{opt}}}} \tag{5}$$

Another metric is the average residual energy in terminal states $s_T$: $E^{\text{res}}(s_T)$.

$$\overline{E^{\text{res}}}_T = \frac{1}{|\mathcal{D}^{\text{test-traj}}|} \sum_{s \in \mathcal{D}^{\text{test-traj}}} (E_{s_T} - E_{s_{\mathbf{opt}}}). \qquad (6)$$

Generally accepted chemical accuracy is 1 kcal/mol(4.184 kJ/mol) [88]. Thus, another important metric is the percentage of conformations for which the residual energy is less than chemical accuracy. We consider optimizations with such residual energies successful:

$$\text{pct}_{\text{success}} = \frac{1}{|\mathcal{D}^{\text{test-traj}}|} \sum_{s \in \mathcal{D}^{\text{test-traj}}} I\left[E^{\text{res}}(s_T) < 1\right]. \qquad (7)$$

Finally, we denote the percentage of diverged (terminal energy is larger than the initial energy or DFT calculation was unsuccessful) optimizations as $\text{pct}_{\text{div}}$.

To measure the difference between two conformations $s$ and $\tilde{s}$ of the same molecule, we use the `GetBestRMSD` in the RDKit package and denote the root-mean-square deviation as $\text{RMSD}(s, \tilde{s})$.

### B.4  Atomization energy

For a molecule $m$, the formation energy $E_{Form}$ is obtained by subtracting atomization energy $E_{Atom}$ from the total DFT energy, where $E_{Atom}(m) = \sum_{atom \in m} E_{atom}$. The quantity $E_{atom}$ is the energy of a system consisting of a single atom. Thus, it depends only on the atom type. This operation, while being just a bias/dispersion correction, seems to be a hard task for end-to-end training of state-of-the-art models in our setup.

## C  Applications for the Drug Discovery

The proposed dataset includes a large amount of Quantum Chemistry (QC) data that is important both for the manual analysis of chemical properties and the training of Neural Network models. The dataset includes:

- **Energies and forces.** The potential energy and interatomic forces are fundamental properties of the atomic system that define the dynamics of the system in an environment. Accurate prediction of the interatomic forces allows to carry out molecular dynamics simulation that are for example employed in alchemical free energy calculations [89]. Calculated binding free energy could serve as criteria for selecting promising ligands [90].
- **Optimization trajectories.** Understanding the local minima of the Potential Energy Surface (conformers) is an important task, as these represent the most likely states for a molecule. Conformers are usually acquired through iterative optimization. We included the optimization trajectories in the dataset to estimate how the predicted forces can be used in iterative optimization.
- **Hamiltonians and overlap matrices.** This data is used in quantum chemistry computational software to calculate important quantum chemical properties: Molecular electrostatic potential (MEP), Löwdin atomic charges, Wiberg bond indices, the restrained electrostatic potential (REsP), various partial charges, and many other [91]. These properties can, for example, be used for manual analysis of chemical reactivity, bioavailability, and blood-brain barrier permeability [92].

In conclusion, our dataset is a reliable source of QC data for commercially available drug-like substances. We believe it will be instrumental in developing models for structure- and ligand-based drug design, docking pose estimation, and other challenging tasks in computational chemistry.

## D  Additional information and benchmarking

Table 9 details the contents of quantum chemistry datasets, showing all information provided in the $\nabla^2$DFT dataset in comparison with other datasets. Figure 4 shows the RMSD between optimized conformations and optimal geometry from DFT optimization. Table 7 shows energy prediction metrics in terms of mean absolute error (MAE, less is better). Table 8 snows similar results for the forces prediction task.

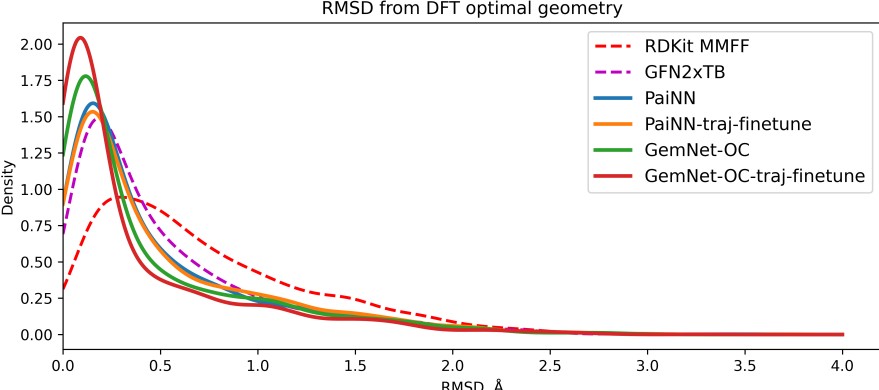

Figure 4: RMSD between optimized conformations and optimal geometry from DFT optimization.

Table 7: Energy prediction metrics: mean absolute error (MAE), less is better.

| Model | MAE for energy prediction, $\times 10^{-2} E_h$ | | | |
|---|---|---|---|---|
| | $\mathcal{D}^{\text{tiny}}$ | $\mathcal{D}^{\text{small}}$ | $\mathcal{D}^{\text{medium}}$ | $\mathcal{D}^{\text{large}}$ |
| **Structure test split** | | | | |
| LR | 4.86 | 4.64 | 4.56 | 4.56 |
| SchNet | 1.17 | 0.90 | 1.10 | 0.31 |
| PaiNN | **0.82** | 0.60 | 0.36 | **0.09** |
| Dimenet++ | 42.84 | 0.56 | **0.21** | **0.09** |
| SchNOrb | 0.83 | **0.47** | 0.39 | 0.39 |
| Graphormer3D | 1.54 | 0.96 | 0.77 | 0.37 |
| GemNet-OC | 2.79 | 0.65 | 0.28 | 0.22 |
| EquiformerV2 | 2.81 | 1.13 | 0.28 | 0.19 |
| eSCN | 1.87 | **0.47** | 0.94 | 0.42 |
| **Scaffolds test split** | | | | |
| LR | 4.37 | 4.18 | 4.12 | 4.15 |
| SchNet | 1.19 | 0.92 | 1.11 | 0.31 |
| PaiNN | **0.86** | 0.61 | 0.36 | 0.09 |
| Dimenet++ | 37.41 | **0.41** | **0.19** | **0.08** |
| SchNOrb | **0.86** | 0.46 | 0.37 | 0.39 |
| Graphormer3D | 1.58 | 0.94 | 0.75 | 0.36 |
| GemNet-OC | 2.59 | 0.59 | 0.27 | 0.23 |
| EquiformerV2 | 2.65 | 1.13 | 0.27 | 0.17 |
| eSCN | 1.87 | 0.47 | 0.92 | 0.42 |
| **Conformations test split** | | | | |
| LR | 3.76 | 3.61 | 3.69 | 3.95 |
| SchNet | 0.56 | 0.63 | 0.88 | 0.28 |
| PaiNN | 0.43 | 0.49 | 0.28 | 0.08 |
| Dimenet++ | 0.42 | **0.10** | **0.09** | **0.07** |
| SchNOrb | **0.37** | 0.26 | 0.27 | 0.36 |
| Graphormer3D$_{Small}$ | 0.99 | 0.67 | 0.58 | 0.39 |
| GemNet-OC | 0.52 | 0.20 | 0.15 | 0.24 |
| EquiformerV2 | 0.45 | 0.23 | 0.24 | 0.16 |
| eSCN | 0.48 | 0.31 | 0.80 | 0.44 |

Table 8: Forces prediction metrics: mean absolute error (MAE), less is better.

| Model | MAE for forces prediction, $\times 10^{-2} E_h/\text{Å}$ | | | |
|---|---|---|---|---|
| | $\mathcal{D}^{\text{tiny}}$ | $\mathcal{D}^{\text{small}}$ | $\mathcal{D}^{\text{medium}}$ | $\mathcal{D}^{\text{large}}$ |
| **Structure test split** | | | | |
| SchNet | 0.44 | 0.37 | 0.41 | 0.16 |
| PaiNN | 0.37 | 0.26 | 0.17 | 0.06 |
| Dimenet++ | 1.31 | 0.20 | 0.13 | 0.07 |
| Graphormer3D | 1.11 | 0.67 | 0.54 | 0.26 |
| GemNet-OC | 0.14 | 0.07 | 0.05 | **0.02** |
| EquiformerV2 | 0.30 | 0.23 | 0.21 | 0.17 |
| eSCN | **0.10** | **0.05** | **0.04** | **0.02** |
| **Scaffolds test split** | | | | |
| SchNet | 0.45 | 0.37 | 0.41 | 0.16 |
| PaiNN | 0.38 | 0.26 | 0.17 | 0.06 |
| Dimenet++ | 1.36 | 0.19 | 0.13 | 0.07 |
| Graphormer3D | 1.13 | 0.68 | 0.55 | 0.26 |
| GemNet-OC | 0.14 | 0.06 | **0.04** | **0.02** |
| EquiformerV2 | 0.31 | 0.23 | 0.21 | 0.17 |
| eSCN | **0.10** | **0.05** | **0.04** | **0.02** |
| **Conformations test split** | | | | |
| SchNet | 0.32 | 0.30 | 0.37 | 0.14 |
| PaiNN | 0.23 | 0.22 | 0.14 | 0.05 |
| Dimenet++ | 0.26 | 0.12 | 0.10 | 0.06 |
| Graphormer3D | 0.82 | 0.54 | 0.45 | 0.23 |
| GemNet-OC | **0.07** | **0.04** | **0.03** | **0.02** |
| EquiformerV2 | 0.16 | 0.15 | 0.16 | 0.13 |
| eSCN | **0.07** | **0.04** | **0.03** | **0.02** |

Table 9: Content of quantum chemistry datasets.

| | |
|---|---|
| $\nabla^2$DFT/$\nabla$DFT | Atom numbers, atom positions, energy, forces, Hamiltonian (Fock matrix), overlap matrix, coefficients matrix, 'DFT FORMATION ENERGY', 'DFT TOTAL ENERGY', 'DFT XC ENERGY', 'DFT NUCLEAR REPULSION ENERGY', 'DFT ONE-ELECTRON ENERGY', 'DFT TWO-ELECTRON ENERGY', 'DFT DIPOLE X', 'DFT DIPOLE Y', 'DFT DIPOLE Z', 'DFT TOTAL DIPOLE', 'DFT ROT CONSTANT A', 'DFT ROT CONSTANT B', 'DFT ROT CONSTANT C', 'DFT HOMO', 'DFT LUMO', 'DFT HOMO-LUMO GAP','DFT ATOMIC ENERGY', Ca/Cb, Da/Db, Fa/Fb, H, S, X, aotoso, epsilon_a/epsilon_b, SCF DIPOLE, doccpi, nmo, 'DISPERSION CORRECTION ENERGY', 'GRID ELECTRONS TOTAL', electric dipole moment, electric quadrupole moment, all moments up order N, electrostatic potential at nuclei, electrostatic potential on grid, electric field on grid, molecular orbital extents, Mulliken atomic charges, Löwdin atomic charges, Wiberg bond indices, Mayer bond indices, natural orbital occupations, Stockholder Atomic Multipoles, Hirshfeld volume ratios |
| QM7 | Coulomb matrices, atomization energies |
| QM7b | 13 properties (e.g. polarizability, HOMO and LUMO eigenvalues, excitation energies) |
| QM7-X | Atomic numbers, atomic positions, RMSD to optimized structure, moment of inertia tensor, total PBE0+MBD energy, total DFTB3+MBD energy, atomization energy, PBE0 energy, MBD energy, TS dispersion energy, nuclear-nuclear repulsion energy, kinetic energy, nuclear-electron attraction, classical coulomb energy, exchange-correlation energy, exchange energy, correlation energy, exact exchange energy, sum of Kohn-Sham eigenvalues, Kohn-Sham eigenvalues, HOMO energy, LUMO energy, HOMO-LUMO gap, scalar dipole moment, Dipole moment, Total quadrupole moment, ionic quadrupole moment, electronic quadrupole moment, molecular C6 coefficient, molecular polarizability, molecular polarizability tensor, total PBE0+MBD atomic forces, PBE0 atomic forces, MBD atomic forces, Hirshfeld volumes, Hirshfeld ratios, Hirshfeld charges, scalar Hirshfeld dipole moments, Hirshfeld dipole moments, Atomic C6 coefficients, Atomic polarizabilities, vdW radii |
| QM9 | DFT + partially G4MP2: rotational constants, dipole moment, isotropic polarizability, HOMO/LUMO/gap energies, electronic spatial extent, zero point vibrational energy, internal energy at 0 K, internal energy at 298.15 K, enthalpy at 298.15 K, free energy at 298.15 K, heat capacity at 298.15 K, Mulliken charges, harmonic vibrational frequencies |
| MultiXC-QM9 | Semi-empirical energies with XTB method of molecules, atomization energies with all basis sets and functionals, DFT energies with TZP basis of molecules and bond lists , index of reactions, reactants, and products, reaction energy for A;B reactions, reaction energy for A-B reactions,DFT energies with SZ basis of the molecules, bond change for reactions and reaction energies, DFT energies with TZP basis of the molecules, xyz files |
| QM1B | Energy, HOMO, LUMO, the number of atomic orbitals, the standard deviation of the energy of the last five iterations, HOMO-LUMO gap |
| QH9 | Hamiltonian matrices |
| GEOM | Degeneracy, total energy, relative energy, Boltzmann weight, conformer weights |
| ANI-1 | Total energies, atomization energies |
| ANI-1x, ANI-1ccx | Atomic positions, atomic numbers, total energy, HF energy, NPNO-CCSD(T), correlation, energy, MP2, correlation, energy, atomic forces, molecular moments, electric moments, atomic charges, atomic, electric, moments, atomic volumes |
| OrbNet Denali | Total energy, charge of the molecule |
| SPICE | Dipole and quadrupole moments; MBIS charges, dipoles, quadrupoles, and octopoles for each atom; Wiberg bond orders; and Mayer bond orders. |
| PubChemQC | Molecular formula, Canonical SMILES, charge, HOMO, LUMO, HOMO-LUMO gap, total dipole moment, orbital energies, number of basis, Mulliken populations, Löwdin populations, molecular weight, InChI strings, multiplicity. |
| Frag20 | SMILES, 3D Structure, formation energy. |
| VQM24 | Stoichiometry, atomic Numbers, Cartesian coordinates, SMILES, InCHI strings, total energies, internal energies, atomization energies, electron-electron energies, exchange correlation energies, dispersion energy, HOMO-LUMO gap, dipole moments, quadrupole moments, octupole moments, hexadecapole moments, rotational constants, vibrational eigen modes, vibrational frequencies, free energy, internal (thermal) energy, enthalpy, zero point vibrational energy, entropy, heat capacities, electrostatic potentials at nuclei, Mulliken charges, MO energies (molden files), wavefunctions (molden files), error bars. |
| QMugs | ChEMBL identifier, conformer identifier, total energy, internal atomic energy, formation energy, total enthalpy, total free energy, dipole, quadrupole, rotational constants, enthalpy, heat capacity, entropy, HOMO energy, LUMO energy, HOMO-LUMO gap, Fermi level, Mulliken partial charges, covalent coordination number, molecular dispersion coefficient, atomic dispersion coefficients, molecular polarizability, Atomic polarizabilities, Wiberg bond orders,total Wiberg bond orders, total energy, total internal atomic energy, formation energy, electrostatic potential, Löwdin partial charges, Mulliken partial charges, rotational constants, dipole, exchange-correlation energy, nuclear repulsion energy, one-electron energy, two-electron energy, HOMO energy, LUMO energy, HOMO-LUMO gap, Mayer bond orders, Wiberg-Löwdin bond orders, total Mayer bond orders, total Wiberg-Löwdin bond orders; alpha density matrix, beta density matrix, alpha orbitals, beta orbitals, atomic-orbital-to-symmetry-orbital transformer, Mayer bond orders, Wiberg-Löwdin bond orders. |

