# Datasheet: $\nabla^2$DFT

Authors: Kuzma Khrabrov, Anton Ber, Artem Tsypin, Egor Rumiantsev, Konstantin Ushenin, Alexander Telepov, Artur Kadurin.
Organization: Artificial Intelligence Research Institute(AIRI).

We documented $\nabla^2$DFT and intended uses through the Datasheets for Datasets framework [Gebru et al., 2021]. The goal of dataset datasheets as outlined by Gebru et al. [2021] is to provide a standardized process for documentating datasets. Gebru et al. [2021] present a list of carefully selected questions that dataset authors should answer. We hope our answers to these questions will facilitate better communication between us (the dataset creators) and future users of $\nabla^2$DFT.

# 1 Motivation

## 1.1 For what purpose was the dataset created?

**Purpose** This dataset was created to benchmark state-of-the-art deep neural networks for quantum chemistry and to perform ML-related studies in quantum chemistry with the widest range of possible problem statements.

## 1.2 Who created this dataset and on behalf of which entity?

The dataset was collected by Artur Kadurin's research group and external co-authors on behalf of the Artificial Intelligence Research Institute (AIRI).

## 1.3 Who funded the creation of the dataset?

The creation and maintenance of the dataset are funded by the Artificial Intelligence Research Institute (AIRI).

# 2    Composition

## 2.1    What do the instances that comprise the dataset represent (e.g. documents, photos, people, countries)?

The instances (rows) of the dataset represent molecular conformations. Each conformation represents spatial arrangement of $N$ atoms of the molecule. The conformations consists of atomic coordinates (positions) $\mathbf{R} \in \mathbb{R}^{N \times 3}$ and atomic numbers $Z \in \mathbb{N}^N$.

## 2.2    How many instances are there in total (of each type, if appropriate)?

The dataset contains ~16M conformations for ~2M molecules. For each molecule there is 1 to 63 different conformations generated through a two-step procedure that involves generating conformations with the ETKDG algorithm [Wang et al., 2020] and clustering them with Butina [Barnard and Downs, 1992] clustering algorithm.

## 2.3    Does the dataset contain all possible instances or is it a sample (not necessarily random) of instances from a larger set?

The $\nabla^2$DFT dataset is derived from the MOSES dataset [Polykovskiy et al., 2020], which comprises 1,936,962 molecules. This dataset includes all molecules from the MOSES dataset, with the exception of 33 molecules that were filtered during the conformation generation process. The MOSES dataset itself is based on the ZINC Clean Leads dataset [Irwin et al., 2012], which contains 4,591,276 molecules.

## 2.4    What data does each instance consist of?

Each row/instance of the $\nabla^2$DFT dataset contains the input and output of a DFT evaluation:

$$f(\text{atom positions}, \text{atom types}) = \text{quantum chemical properties}.$$

Each row also includes the SMILES string representation of the molecule from the ZINC Clean Leads dataset.

The $\nabla^2$DFT dataset is stored in five different formats:

1. **An archive of .xyz files with atom positions and types** The archive is available at https://a002dlils-kadurin-nabladft.obs.ru-moscow-1.hc.sbercloud.ru/data/nablaDFTv2/conformers_archive_v2.tar

2. **SQLLite ASE databases** with the following features:

- **smiles**: the SMILES string taken from MOSES. There are up to 63 rows (i.e. conformers) with the same SMILES string;

- **atoms**: string representing the atom symbols of the molecule, e.g. "COOH";

- **positions**: Cartesian coordinates $\mathbf{R} \in \mathbb{R}^{N \times 3}$ of atoms in the molecule (Angstrom);

- **energy**: formation energy of the molecule calculated with Psi4 (Hartree);

- **forces**: interatomic forces $\mathbf{F} \in \mathbb{R}^{N \times 3}$ (Hartree/Angstroem);

- **iteration**: optional field that indicates the optimization step number (only for optimization trajectories);

- **moses_id**: integer ID of a molecule in MOSES dataset;

- **conformation_id**: integer ID of a conformation.

The full list of ASE databases is available at https://github.com/AIRI-Institute/nablaDFT/blob/main/nablaDFT/links/energy_databases.json

3. **Hamiltonian databases for train/test splits with the following features**

- **atoms**: string representing the atom symbols of the molecule, e.g. "COOH";

- **positions**: Cartesian coordinates $\mathbf{R} \in \mathbb{R}^{N \times 3}$ of atoms in the molecule (Bohr);

- **energy**: formation energy of the molecule calculated with Psi4 (Hartree);

- **forces**: interatomic forces $\mathbf{F} \in \mathbb{R}^{N \times 3}$ (Hartree/Bohr);

- **Hamiltonian**: Hamiltonian matrix $\mathbf{H} \in \mathbb{R}^{M \times M}$, where $M$ is the size of the electronic orbital basis (Hartree);

- **overlap**: overlap matrix $\mathbf{S} \in \mathbb{R}^{M \times M}$, where $M$ is the size of the electronic orbital basis (no unit);

- **moses_id**: integer ID of a molecule in MOSES dataset;

- **conformation_id**: integer ID of a conformation.

The full list of Hamiltonian databases is available at https://github.com/AIRI-Institute/nablaDFT/blob/main/nablaDFT/links/hamiltonian_databases.json

4. **Tar archives of Psi4 raw wavefunction files.** The full list is available at https://github.com/AIRI-Institute/nablaDFT/blob/main/nablaDFT/links/nablaDFT_psi4wfn_links.txt.

5. **Gzipped csv files with base and trajectory conformations.** Full summary file with the following features:

- **smiles**: the SMILES string taken from MOSES;

- `splits`: names of splits that contain this conformation;

- `MOSES id`: integer ID of a molecule in MOSES dataset;

- `conformation id`: integer ID of a conformation;

- `archive name`: name of wavefunctions acrhive that contains the wavefunction object for this conformations;

- `DFT TOTAL ENERGY`: total energy of the system (Hartree);

- `DFT XC ENERGY`: energy of the exchange-correlation functional (Hartree);

- `DFT NUCLEAR REPULSION ENERGY`: energy of the repulsive nucleus interaction (Hartree);

- `DFT ONE-ELECTRON ENERGY` the full energy of non-interacting electrons (Hartree);

- `DFT TWO-ELECTRON ENERGY` the full energy of interacting pairs of electrons (Hartree);

- `DFT DIPOLE X`: component of molecule's dipole along the X axis (Debye);

- `DFT DIPOLE Y` component of molecule's dipole along the Y axis (Debye);

- `DFT DIPOLE Z` component of molecule's dipole along the Z axis (Debye);

- `DFT TOTAL DIPOLE` magnitude of molecular dipole (L2-norm) (Debye);

- `DFT ROT CONSTANT A` the rotational constant A (1/cm);

- `DFT ROT CONSTANT B` the rotational constant B (1/cm);

- `DFT ROT CONSTANT C` the rotational constant C (1/cm);

- `DFT HOMO`: energy of the highest occupied orbital (Hartree);

- `DFT LUMO` energy of the lowest unoccupied orbital (Hartree);

- `DFT HOMO-LUMO GAP`: is the energy gap between the highest occupied and lowest unoccupied molecular orbital (Hartree);

- `DFT ATOMIC ENERGY`: total energy of single atoms (Hartree);

- `DFT FORMATION ENERGY`: is a total energy minus atomic energy (Hartree).

Trajectories summary file with the following features:

- `MOSES id`: Integer ID of a molecule in MOSES dataset

- `conformation id`: Integer id of a conformation

- `iteration`: Integer id of an iteration of the optimization

- `splits`: The string that contains the names of splits, which contain the intial conformation of an optimization trajectory

- `GRAD NORM`: Interatomic forces L2 norm.

- `DFT FORMATION ENERGY`: energy of single atoms.

## 2.5 Is there a label or target associated with each instance?

See Section 2.4. Each instance has several potential labels/prediction targets: i.e. (energy, forces, Hamiltonian, and overlap).

## 2.6 Is any information missing from individual instances?

For optimization trajectories only energies and forces are present. The forces have currently been computed only for 40% of the dataset.

## 2.7 Are relationships between individual instances made explicit?

Each instance contains MOSES ID and CONFORMATION ID. Instances of optimization trajectories also contain iteration numbers. MOSES ID is a unique index of molecules in SMILES form. This index is related to the ID of the molecule in the MOSES database. Several instances can share a common MOSES ID. This means that these instances are different conformations of the same molecule. All conformations of the same molecule have different CONFORMATION IDs. The iteration number is available for conformations in optimization trajectories. It denotes the iteration number in the optimization trajectory of conformation CONFORMATION ID of molecule MOSES ID. Thus, the index system looks like a tree data structure or folders in the file system. MOSES ID denotes top-level nodes of the tree. CONFORMATION ID denotes leaves or intermediate nodes of the tree. The index of iterations in the optimization trajectory denotes the leaves of the tree. There are no relation links between instances like in relation or graph databases.

## 2.8 Are there recommended data splits?

Yes, we propose 5 train splits with various sizes and 3 families of test splits: (1) conformation test sets are designed to test the ability of the models to generalize to unseen *geometries* of molecules; (2) structure and scaffold test sets, to unseen *molecules*. The conformation test set is the easiest and the scaffold test set is the most challenging as it contains unseen molecular fragments. More details are contained in the paper.

## 2.9 Are there any errors, sources of noise, or redundancies in the dataset?

PSI4 and RDKit have limitations and cannot ensure completely accurate computations due to inherent constraints of the Linear Combination of Atomic Orbitals (LCAO) method and the software implementations themselves. Outlier conformations that are not physically realistic are excluded. Several optimization trajectories contain unrealistic energy values (final energy is larger than the initial energy) due to the nature of the optimization algorithm. Please, take it into account when evaluating new methods on this data. Density Functional Theory (DFT) does not replicate real chemical experiments with perfect chemical accuracy (approximately 1 kcal/mol). Consequently, the dataset inherits biases associated with the theoretical level, the basis set, and the chosen exchange-correlation functional.

## 2.10 Is the dataset self-contained, or does it link to or otherwise rely on external resources (e.g. websites, tweets, other datasets)?

Yes, the dataset is self-contained. Additional information about molecules can be found in MOSES and ZINC Clean Lead datasets. The MOSES dataset is available under the MIT License, and the ZINC Clean Leads license and terms of services are available on the official site of the ZINC database.

## 2.11 Does the dataset contain data that might be considered confidential (e.g. data that is protected by legal privilege or by doctor-patient confidentiality, data that includes the content of individuals' non-public communications)?

No. The data contains molecules that are not considered confidential.

## 2.12 Does the dataset contain data that, if viewed directly, might be offensive, insulting, threatening, or might otherwise cause anxiety?

No.

## 2.13 Additional comments

The preprocessed part of $\nabla^2$DFT (15 Tb) includes Hamiltonians, forces, energies, and 3D structures of the molecule. The preprocessed part of $\nabla^2$DFT is divided into several data splits. To get familiar with the dataset, we recommend starting with `train_tiny` and `test_tiny_conformers` splits. The full dataset (220 Tb) includes Psi4 wavefunction objects. The meta information

in `summary.csv` can be used to compose a download list for a specific set of molecules.

# 3   Collection

## 3.1   How was the data associated with each instance acquired?

The $\nabla^2$DFT dataset is derived from the MOSES dataset using numerical methods implemented in RDKit and Psi4 software. The MOSES dataset is based on the ZINC Clean Leads dataset. The filtration of ZINC Clean Leads that results in the MOSES dataset is described in [Polykovskiy et al., 2020]. Additional filtraition of MOSES dataset is described in Section 3 of the main paper. Conformations in the $\nabla^2$DFT are generated using RDKit and Psi4 software, as described in Section 4.

## 3.2   What mechanisms or procedures were used to collect the data (e.g. hardware apparatus or sensor, manual human curation, software program, software API)?

Initial chemical information for the dataset was acquired from the MOSES dataset. Each instance of $\nabla^2$DFT is generated using RDKit and Psi4 software, as described in Section 4.

## 3.3   If the dataset is a sample from a larger set, what was the sampling strategy (e.g. deterministic, probabilistic with specific sampling probabilities)?

$\nabla^2$DFT contains molecules from MOSES dataset. We applied a multi-step filtration protocol to ensure the validity of the provided QC computations. In total we filtered 33 molecules from the MOSES dataset.

## 3.4   Who was involved in the data collection process (e.g. students, crowd workers, contractors), and how were they compensated (e.g. how much were crowd workers paid)?

Data collection and processing were performed by permanent employees of the organization, which is mentioned in the header of the main article.

## 3.5   Over what timeframe was the data collected?

Original chemical information for the dataset was downloaded from the MOSES dataset on the 1st of November 2021. All the calculations performed to collect the dataset require ~60 CPU years, which corresponds to ~6 months of wall-time

on the available computational facility. The dataset was gradually increased and improved until the May of 2024.

## 3.6 Were any ethical review processes conducted (e.g. by an institutional review board)?

No.

# 4 Preprocessing / Cleaning / Labeling

## 4.1 Was any preprocessing/cleaning/labeling of the data done (e.g. discretization or bucketing, tokenization, part-of-speech tagging, SIFT feature extraction, removal of instances, processing of missing values)?

The $\nabla^2$DFT dataset is based on the MOSES dataset, and the last one is based on the ZINC Clean Leads collection. The original database contains 4,591,276 molecules in total. MOSES instances were filtered with the following criteria: (1) molecular weight in the range from 250 to 350 Daltons; (2) the number of rotatable bonds not greater than 7; (3) and XlogP less than or equal to 3.5; (4) molecules do not contain charged atoms; (5) molecules do not contain atoms besides C, N, S, O, F, Cl, Br, H; (6) molecules do not contain cycles longer than 8 atoms; (7) molecules pass medicinal chemistry filters (MCFs) and PAINS filters; More details can be found in [Polykovskiy et al., 2020].

We applied a multi-step filtration protocol to ensure the validity of the provided QC computations. We filtered 31 molecules from the MOSES dataset where the conformation generation procedure could not produce valid conformations. Then, we filtered samples with anomalous values of QC properties: *('DTF TOTAL ENERGY' > 0)*, *('DFT TOTAL DIPOLE' < 20)*, or *('DFT FORMATION ENERGY' < 0)*. We excluded 29 conformations and discarded 2 more molecules, totaling 33.

Molecular properties were calculated with the Kohn-Sham theory with the linear combination of atomic orbitals (LCAO) numerical approach using the Psi4 software package (version 1.5). We use $\omega$B97X-D basis set and def2-SVP level of theory. Default Psi4 parameters were used for DFT computations, i.e. Lebedev-Treutler grid with a Treutler partition of the atomic weights, 75 radial points, and 302 spherical points, the criterion for the SCF cycle termination was the convergence of energy and density up to $10^{-6}$ threshold, integral calculation threshold was $10^{-12}$.

## 4.2 Was the "raw" data saved in addition to the preprocessed/cleaned/labeled data (e.g. to support unanticipated future uses)?

The $\nabla^2$DFT contains full psi4 wavefunction objects, that include all information about the computation process. All links are available in the official GitHub repository: https://github.com/AIRI-Institute/nablaDFT.

## 4.3 Is the software used to preprocess/clean/label the instances available?

Software for computational chemistry:

- Psi4, version 1.5 [Smith et al., 2020];

- RDKit, version 2022.03.1 [Landrum et al., 2022].

Software for utility operations:

- Python, version 3.10
  https://www.python.org/downloads/release/python-31014/;

- pandas, https://pandas.pydata.org/;

- matplotlib, https://matplotlib.org/;

- seaborn, https://seaborn.pydata.org/.

# 5 Uses

## 5.1 Has the dataset been used for any tasks already?

Our dataset is used as a benchmark for state-of-the-art quantum chemistry models. We tested 8 models that predict energy from the 3D structure of the molecule (atom positions and types), 2 models that predict energy, and Hamiltonians (overlap and Fock matrix).

## 5.2 Is there a repository that links to any or all papers or systems that use the dataset?

The current version of the dataset has been recently used in [Tsypin et al., 2024]. Our dataset is the extended and improved version of the nablaDFT dataset [Khrabrov et al., 2022]. The full list of citations for the first version is available in scholar.

## 5.3 What (other) tasks could the dataset be used for?

The $\nabla^2$DFT dataset aims to cover as many ML-based problem statements in tasks related to druglike molecules. $\nabla^2$DFT can be used to predict any standard

molecular properties directly from SMILES, 3D structure of molecule, density or Hamiltonian matrix. The full $\nabla^2$DFT dataset includes Psi4 wavefunction objects. Thus, data can be loaded into the Psi4 software and any non-standard property of the molecule can be computed. The dataset can also be used to help train the wavefunction predicting models.

## 5.4 Is there anything about the composition of the dataset or the way it was collected and preprocessed/cleaned/labeled that might impact future uses?

As the dataset is based on MOSES [Polykovskiy et al., 2020], it only contains conformations of small druglike molecules. This dataset can be used for pre-training of NNPs or Hamiltonian predicting models for more complex domains such as protein-ligand interactions and solutions.

## 5.5 Are there tasks for which the dataset should not be used?

The $\nabla^2$DFT dataset is not suitable for studying the properties of materials, solvated molecules or protein-ligand pairs (important for ML applications in drug design). It lacks charged and open-shell systems, nano-particles, nanotubes, big rings, and other non-drug-like structures. So the generalization of models trained solely on $\nabla^2$DFT to these domains is not expected. Moreover, $\nabla^2$DFT is unsuitable for inorganic chemistry and for ML-based studies of long-range and non-covalent interactions.

# 6 Distribution

## 6.1 Will the dataset be distributed to third parties outside of the entity (e.g. company, institution, organization) on behalf of which the dataset was created?

Yes, the dataset is in open access.

## 6.2 How will the dataset will be distributed (e.g. tarball on website, API, GitHub)?

The official repository of the dataset is: https://github.com/AIRI-Institute/nablaDFT. This repository includes links to the data. The dataset is distributed by cloud.ru via Evolution Object Storage. Cloud.ru is an independent cloud provider, and Evolution Object Storage is a service similar to Amazon S3 Cloud Object Storage.

### 6.3   When will the dataset be distributed?

The first version of nablaDFT has been available since the 26th of August 2022. $\nabla^2$DFT is available since the 5th of June 2024.

### 6.4   Will the dataset be distributed under a copyright or other intellectual property (IP) license, and/or under applicable terms of use (ToU)?

The dataset is distributed under the MIT License.

### 6.5   Have any third parties imposed IP-based or other restrictions on the data associated with the instances?

No.

### 6.6   Do any export controls or other regulatory restrictions apply to the dataset or to individual instances?

No.

## 7   Maintenance

### 7.1   Who is supporting/hosting/maintaining the dataset?

**Support**: The authors will provide support for $\nabla^2$DFT through GitHub.

**Hosting**: Dataset is hosted in the [cloud.ru](cloud.ru) cloud platform and is available worldwide. The hosting expenses are covered by AIRI.

**Maintenance**: The maintenance of dataset updates and corrections will be facilitated through the GitHub repository [$\nabla^2$DFT](). We invite users to provide feedback and submit questions and bug reports as GitHub Issues. We will document corrections and updates through the CHANGELOG and provide versioned releases for any major updates.

### 7.2   How can the owner/curator/manager of the dataset be contacted (e.g. email address)?

We encourage public discussion via GitHub issues: [https://github.com/AIRI-Institute/nablaDFT/issues](https://github.com/AIRI-Institute/nablaDFT/issues). Otherwise, the authors of dataset can be contacted via official [mailto:nablaDFT@airi.net](mailto:nablaDFT@airi.net). The curator of the dataset is Kuzma Khrabrov [mailto:khrabrov@airi.net](mailto:khrabrov@airi.net). The manager of the dataset is Artur Kadurin [mailto:kadurin@airi.net](mailto:kadurin@airi.net).

## 7.3 Is there an erratum? If so, please provide a link or other access point.

Currently no. A list of possible misprints and errors will be added in the main repository of the dataset https://github.com/AIRI-Institute/nablaDFT if any issues are found.

## 7.4 Will the dataset be updated (e.g. to correct labeling errors, add new instances, or delete instances)? If so, please describe how often, by whom, and how updates will be communicated to users (e.g. mailing list, GitHub)?

The support will be performed by Kuzma Khrabrov and Anton Ber. The research team plans to continuously answer GitHub issues, and update tutorials, documentation, statistical information, images, and meta-information about the dataset.

## 7.5 If the dataset relates to people, are there applicable limits on the retention of the data associated with the instances (e.g. were individuals in question told that their data would be retained for a fixed period of time and then deleted)? If so, please describe these limits and explain how they will be enforced.

$\nabla^2$DFT contains molecules and does not relate to people.

## 7.6 Will older versions of the dataset continue to be supported/hosted/maintained? If so, please describe how. If not, please describe how its obsolescence will be communicated to dataset consumers

The first version of the nablaDFT dataset [Khrabrov et al., 2022] is a subset of $\nabla^2$DFT. We do not plan to support old versions of nablaDFT, but all the data and associated is available. If we create a significantly advanced version of the dataset in the future, we will also replace $\nabla^2$DFT with an extended version. The replacement will be announced in the official GitHub page https://github.com/AIRI-Institute/nablaDFT.

**7.7 If others want to extend/augment/build on/contribute to the dataset, is there a mechanism for them to do so? If so, please provide a description. Will these contributions be validated/verified? If so, please describe how. If not, why not? Is there a process for communicating/distributing these contributions to other users? If so, please provide a description.**

Yes, we accept pull requests to $\nabla^2$DFT.