# OpenReview forum: "$\nabla^2$DFT: A Universal Quantum Chemistry Dataset of Drug-Like Molecules and a Benchmark for Neural Network Potentials"
_NeurIPS.cc/2024/Datasets_and_Benchmarks_Track — NeurIPS 2024 Track Datasets and Benchmarks Poster_

### Official Review · Reviewer_taPs · 2024-07-21
**\nabla^2 DFT: A Universal Quantum Chemistry Dataset of Drug-Like Molecules and a Benchmark for Neural Network Potentials**

**Rating:** 6
**Confidence:** 3
**Clarity:** Overall this paper is well written.

**Review:**

**Strength**:
- A universal dataset that includes more than 30 QC properties such as energies, forces, Hamiltonians and overlap matrices, wavefunction objects, and optimization trajectories for druglike molecules
- A comprehensive benchmark for evaluating quantum chemistry models, encompassing tasks such as Hamiltonian prediction, energy and force prediction, and conformational optimization
- An extendable framework2 that contains adaptations of 10 quantum chemistry models together with reported benchmark metrics and checkpoints.

**Weakness**:
- For the geometry optimization, more recent work should be included in the benchmark, such as equiformerV2.
- Comparing RMSD between final conformation and ground truth conformation may not be very informative. It's better to compare the DFT energy of those two conformers to evaluate which one is more stable.
-  It is better to provide a jupyter notebook to demonstrate end-to-end on how to load data, process data, train the model, and evaluate the performance, as this will be very useful for users to quickly use the dataset.

**Strengths:**

Please refer to the strength in **Review**

**Additional Feedback:**

N/A

**Correctness:**

Please refer to the weakness part in **Review** for the comments on geometry optimization.

**Documentation:**

Documentation is sufficient.

**Limitations:**

Limitation is well discussed.

**Opportunities For Improvement:**

Please refer to the weakness in **Review**

**Relation To Prior Work:**

Relation to prior work is clearly discussed.

**Summary And Contributions:**

This paper proposes a new dataset and benchmark that is based on nablaDFT. It contains twice as much molecular structures, three times more conformations, new data types and tasks, and state-of-the-art models. The dataset includes energies, forces, 17 molecular properties, Hamiltonian and overlap matrices, and a wavefunction object. They also introduce a novel benchmark for evaluating NNPs in molecular property prediction, Hamiltonian prediction, and conformational optimization tasks.

---

> ### Author Rebuttal · Authors · 2024-08-13
>
> Thank you for your useful comments!
>
> ## Weaknesses:
>
> - Weakness 1
>
> > For the geometry optimization, more recent work should be included in the benchmark, such as equiformerV2.
>
> We also believe that it is important to track new models and include them in the benchmark. We are going to continue updating nabla2DFT in the future as well. Regarding EquiformerV2, we already reimplemented and evaluated it within our framework together with eSCN and GemNet-OC(please see table below); we will add these results to the manuscript revision.
>
> | Metric                                | Model              | $\mathcal{D^{\text{tiny}}}$| $\mathcal{D^{\text{small}}}$ | $\mathcal{D^{\text{medium}}}$ | $\mathcal{D^{\text{large}}}$
> |--|--|--|--|--|--|
> | $\overline{\operatorname{pct}}_T$ | GemNet-OC | 69.04       | 85.57 | 92.42  | 100.06 |
> |  | GemNet-OC-finetune |  |    |    | 100.005 |
> |   | eSCN | 76.83 | 85.94 | 89.34        | 97.27       |
> |   | EquiformerV2 | 64.41 | 76.11       | 75.24        | 86.10       |
> | $\operatorname{pct}_{\text{success}}$ | GemNet-OC          | 0.91        | 10.42       | 30.94        | 90.71       |
> |  | GemNet-OC-finetune |  |   |  | 94.55 |
> |  | eSCN| 11.49  | 19.23       | 25.39        | 53.38       |
> |  | EquiformerV2| 6.90        | 12.62       | 16.38        | 32.01       |
> | $\operatorname{pct}_{\text{div}}$     | GemNet-OC          | 11.55       | 0.75        | 0.6          | 0.4         |
> |   | GemNet-OC-finetune |  |  |  | 0 |
> |   | eSCN | 59.1 | 27.7 | 11.00  | 0.8 |
> |    | EquiformerV2  | 92.75 | 84.55 | 84.75| 76.1|
>
> - Weakness 2
>
> >Comparing RMSD between final conformation and ground truth conformation may not be very informative. It's better to compare the DFT energy of those two conformers to evaluate which one is more stable.
>
> We agree that the energies are more informative than RMSD in our case. In Tables 4 and 5 on page 9, we list $\operatorname{pct}$ metrics, which measure the difference between the energy of the DFT-optimal conformation and the energy of the NNP-optimal one. Please see formulas 5-7 in Appendix B.3.

---

> > ### Author Rebuttal · Authors · 2024-08-13
> >
> > - Weakness 3
> >
> > > It is better to provide a jupyter notebook to demonstrate end-to-end on how to load data, process data, train the model, and evaluate the performance, as this will be very useful for users to quickly use the dataset.
> >
> > We agree that such jupyter notebooks are very important for the community and could serve as an entrypoint to the field of NNP for quantum chemistry tasks. We prepared tutorials for:
> >
> > - Basic data manipulation: https://github.com/AIRI-Institute/nablaDFT/blob/main/examples/0a_basic_access.ipynb
> >
> > - Datasets meta-information: https://github.com/AIRI-Institute/nablaDFT/blob/main/examples/1a_meta_information.ipynb
> >
> > - Train pipeline and pre-trained model test for models based on PyTorch Geometric: https://github.com/AIRI-Institute/nablaDFT/blob/main/examples/GemNet-OC_example.ipynb
> >
> > - Train pipeline and pre-trained model test for models based on SchNetPack: https://github.com/AIRI-Institute/nablaDFT/blob/main/examples/PAINN_example.ipynb
> >
> > - Prediction pipeline example: https://github.com/AIRI-Institute/nablaDFT/blob/main/examples/Inference%20example.ipynb

---

### Official Review · Reviewer_VPVC · 2024-07-25
**Large scale quantum chemistry dataset**

**Rating:** 8
**Confidence:** 4
**Clarity:** The paper is well structured and easy…

**Review:**

* The manuscript is well written and provides clear contrasting comparisons with related datasets and benchmarks.
* The benchmark tasks are well motivated by real-world requirements.

**Strengths:**

* The manuscript is well written
* provides clear contrasting comparisons with related datasets and benchmarks.
* The benchmark tasks are well motivated by real-world requirements.

**Additional Feedback:**

A lot of thought and design went into the conformation generation but I wondered if they saw any drawback or problems with using the RDKit conformers directly?

typo at end of line 182: DTF should be DFT

**Correctness:**

My only concern was the lack of error bars.  The variation between different random initializations can be helpful to place in context the comparisons between different model architectures. Is this something the authors are planning on adding?

**Documentation:**

The github repo is available, my only comment would be to consider introducing software interfaces to make it easier to use the datasets within pytorch.

**Ethics:**

No ethics concerns.

**Limitations:**

Limitations are well addressed

**Opportunities For Improvement:**

On the code repo: it would be helpful to integrate the dataset into PyTorch Geometric, as I recall they have automated downloading the dataset locally so this could make the examples a bit easier to get into.

**Relation To Prior Work:**

Yes they have an extensive written section comparing this dataset to prior contributions in the field as well as a summary table.

**Summary And Contributions:**

The paper introduces an update to the nablaDFT dataset that contains more structures and conformations as well as introduces additional tasks and baseline models for comparison.  This is an important research direction and I commend the authors for making this dataset and accompanying models available to the community.

Both Hamiltonian prediction and NNP evaluation are agreed as essential components for accelerating progress in computational chemistry (especially applied to drug discovery).  In particular the selection of the molecular structures are well motivated for solving key problems in small molecule drug discovery programs which have previously been either limited to small datasets of high-accuracy simulations or large datasets with lower-accuracy.

---

> ### Author Rebuttal · Authors · 2024-08-13
>
> Thank you for your useful comments!
>
> - Question 1
>
> >  On the code repo: it would be helpful to integrate the dataset into PyTorch Geometric, as I recall they have automated downloading the dataset locally so this could make the examples a bit easier to get into.
>
> We agree that various interface implementations make the dataset more accessible. Currently, PyTorch Geometric interfaces are implemented for both energy and Hamiltonian datasets with automated download for chosen split, so there is no need for users to manually download the dataset. Please take a look at the repository: https://github.com/AIRI-Institute/nablaDFT/blob/main/nablaDFT/dataset/pyg_datasets.py
> As for the interface inside the PyTorch Geometric framework, we will check opportunities to contribute to our implementation.
>
> - Question 2
>
> > My only concern was the lack of error bars. The variation between different random initializations can be helpful to place in context the comparisons between different model architectures. Is this something the authors are planning on adding?
>
> We agree that it is essential to measure the performance of the NNs with different random initializations. In this study, we used only one random seed due to the limited computational resources, e.g., for the smallest data splits, approximately two days of computations on 6 V100 GPUs were needed to train EquiformerV2 and GemNet-OC. We plan to add more random seeds in the future.
>
> - Question 3
>
> > A lot of thought and design went into the conformation generation but I wondered if they saw any drawback or problems with using the RDKit conformers directly?
>
> We emphasize that the conformers were indeed generated by the RDKit, and the additional work was only done to filter the data, so the DFT computations will become computationally feasible. The EDKTG method from RDKit failed to generate any conformation for 31 molecules from 1.9M, while for the other ones, at least one conformation with a reasonable DFT energy was generated. The only non-RDKit-generated conformers in the dataset are the ones in the optimization trajectories.
>
> - Question 4
>
> > typo at end of line 182: DTF should be DFT
>
> Thank you, we will fix it!

---

### Official Review · Reviewer_qiHs · 2024-07-26
**Impressive dataset and baselines**

**Rating:** 7
**Confidence:** 3
**Correctness:** Everything seems correct.
**Clarity:** The paper is very clear.

**Review:**

The paper presents a dataset with benchamark models, evaluations and weights for atomic structure inspired by previous datasets (MOSES, ZINNC21 and nableDFT).

The dataset can be used to bechmark new ML Schroedinger solvers, or SE predictors or neural potentials.

It is difficult to evaluate the quality of the dataset (convergence of the DFT), but the dataset seems quite comprehensive.

The only ciritc is what it adds to nablaDFT and QH9, a part from adding additional confirmations and atom types.

**Strengths:**

Extenisive dataset, includiing models and trained models.

**Additional Feedback:**

The paper claims that the dataset is for drug applications, it would be nice if this claim could be expanded, showing examples of application in drug design this dataset can be used for. If space is limited, in the annex.

**Documentation:**

website is very well documented. License is also provided.

**Ethics:**

No ethical consideration seems to apply since these are all generated data.

**Limitations:**

The authors clearly and accurately describe the limitations of their work.

**Opportunities For Improvement:**

It would have been nice if the autors would have added more atom types.

**Relation To Prior Work:**

The authors show a clear connection with other dataset/benchmarks in Table 1.

**Summary And Contributions:**

The paper presents an extension of a previous published dataset of DFT computation on rich atom types (8 types).

The authors increased the number of structures and included additional atom types.

The dataset can be used to traine Hamiltonian prediction model (predition of the solution of the SE) or Energy and Forces (Neural Potential).

The authors provide dataset, models and trained models.

In addition the autors provide optimized trajectories (low energy).

Total storage is 220Tb. Impressive.

---

> ### Author Rebuttal · Authors · 2024-08-14
>
> Thank you for your useful comments!
>
> ## Questions and suggestions
>
> - Question 1
>
> >  It would have been nice if the authors would have added more atom types.
>
> Thank you for the suggestion! We hope to add more atom types and charged atoms in the future. Still, we emphasize the fact that the molecules set we work with at the moment is based on the MOSES dataset, which has a fixed set of atom types.
>
> - Question 2
>
> > The paper claims that the dataset is for drug applications, it would be nice if this claim could be expanded, showing examples of application in drug design this dataset can be used for. If space is limited, in the annex.
>
> Thank you for the questions, they highlight the importance of our dataset for future studies and build over our work.
> Our dataset includes Hamiltonians and overlap matrices. This data is used in quantum chemistry computational software to solve several downstream tasks such as computation of one-electron quantum chemistry properties: Molecular electrostatic potential (MEP), Löwdin atomic charges, Wiberg bond indices, various partial charges, etc. [1] These properties can be used for manual analysis of chemical reactivity, bioavailability, hematoencephalic barrier permeability, etc. [2,3] Neural networks that predict Hamiltonians can replace numerical quantum chemistry software in tasks of semi-automatic analysis of chemical substances and drug-like molecules.
>
> The additional importance of Hamiltonian/Overlap matrix data is the calculation of partial charges for docking problems, such as Restrained Electrostatic Potential (REsP). It makes our dataset a good source of reliable information about drug-like substances to develop new neural network architectures for small molecule generation, docking prediction [4], and target-based drug design.
>
> DFT energies and forces from the dataset allow one to train Neural Potentials, which can accelerate Geometry Optimization and Molecular Dynamics. The latter are actively applied in the field of Drug Discovery  [8].
>
> Also, we note that nabla2DFT is based on the MOSES dataset. This dataset consists of commercially available drug-like substances and is used in several studies related to chemoinformatics and the design of new drugs [5]. Incorporating quantum chemistry data into the MOSES dataset provides a reliable resource for enhancing and reapplying drug discovery research using MOSES such as [6-7].
>
> We expect that such datasets may be used to pretrain Foundational models for quantum chemistry problems like this is done in NLP and CV domains of deep learning.
>
>
>
> [1] https://psicode.org/psi4manual/master/oeprop.html
>
> [2] Suresh, Cherumuttathu H., Geetha S. Remya, and Puthannur K. Anjalikrishna. "Molecular electrostatic potential analysis: A powerful tool to interpret and predict chemical reactivity." Wiley Interdisciplinary Reviews: Computational Molecular Science 12.5 (2022): e1601.
>
> [3] Veber, Daniel F., et al. "Molecular properties that influence the oral bioavailability of drug candidates." Journal of medicinal chemistry 45.12 (2002): 2615-2623.
>
> [4] Macip, Guillem, et al. "Haste makes waste: a critical review of docking‐based virtual screening in drug repurposing for SARS‐CoV‐2 main protease (M‐pro) inhibition." Medicinal Research Reviews 42.2 (2022): 744-769.
>
> [5] Polykovskiy, Daniil, et al. "Molecular sets (MOSES): a benchmarking platform for molecular generation models." Frontiers in pharmacology 11 (2020): 565644.
>
> [6] Zhavoronkov, Alex, et al. "Deep learning enables rapid identification of potent DDR1 kinase inhibitors." Nature biotechnology 37.9 (2019): 1038-1040.
>
> [7] Jiménez-Luna, José, et al. "Artificial intelligence in drug discovery: recent advances and future perspectives." Expert opinion on drug discovery 16.9 (2021): 949-959.
>
> [8] Durrant, Jacob D., and J. Andrew McCammon. "Molecular dynamics simulations and drug discovery." BMC biology 9 (2011): 1-9.
>
> [9] Zhang, Meng, Koki Hibi, and Junya Inoue. "GPU-accelerated artificial neural network potential for molecular dynamics simulation." Computer Physics Communications 285 (2023): 108655.
>
> [10] Li, Zijie, et al. "Graph neural networks accelerated molecular dynamics." The Journal of Chemical Physics 156.14 (2022).

---

### Official Review · Reviewer_EvHy · 2024-07-29
**Extension of a large scale DFT dataset using featured with RDKit geometries featured with Hamiltonian matrix targets**

**Rating:** 7
**Confidence:** 3
**Clarity:** The paper is well written.

**Review:**

See below.

**Strengths:**

- The paper is well written and contains a good review for previous datasets.
- The benchmark is comprehensive.
- The conformation optimization experiment is interesting, particularly the models seems to perform well when only trained with RDKit geometries.

**Additional Feedback:**

- Compared to the Fock matrix, the overlap matrix can be quite easily computed with quantum chemistry software. Is it necessary to train ML models to predict the overlap matrix?

- The results of geometry optimization in table 4 & 5 are compelling, even when the models are only trained with RDKit geometries. Do the authors observe that the optimization is good enough without finetuning on DFT trajectories?

- There appears to be some downloading issues raised in the Github page (e.g., downloading a large trunk of data in serial appears to be difficult). Does the authors plan to address these?

**Correctness:**

The dataset is constructed in a sound way. The performance of QHNet seems to be less consistent with prior publications.

**Documentation:**

There are sufficient details.

**Ethics:**

No ethical concerns are raised.

**Limitations:**

The authors addressed the limitations adequately.

**Opportunities For Improvement:**

- The reported results for QHNet is much worse compared to PhisNet. However, in the QHNet paper [1] the results are comparable. What are the causes for this difference?

- On the Github the download link for the full dataset is available for the energy dataset but not for the Hamiltonian dataset. Does the author plan to release the full Hamiltonian dataset?

[1] Yu, Haiyang, et al. "Efficient and equivariant graph networks for predicting quantum Hamiltonian." International Conference on Machine Learning. PMLR, 2023.

**Relation To Prior Work:**

The new version contains more data and includes more experimental benchmarking.

**Summary And Contributions:**

This is an extension to the nablaDFT dataset. More data is included. More benchmark results on energy / force prediction and Hamiltonian matrix prediction are provide. This version additionally provide a smaller amount of DFT optimization trajectories. The authors also tested the ability of geometry optimization from their trained models.

---

> ### Author Rebuttal · Authors · 2024-08-14
>
> Thank you for your useful comments!
>
> ## Questions and suggestions
>
> - Question 1
>
> > The reported results for QHNet is much worse compared to PhisNet. However, in the QHNet paper [1] the results are comparable. What are the causes for this difference?
>
> Thank you for this question. It highlights the importance of our benchmark for the Hamiltonian prediction task. QHNet results were reported in the original work for the MD17 dataset and PBE/def2-SVP level of theory. $\nabla^2$DFT includes F, S, Cl, and Br in addition to atoms presented in MD17. These atoms in the def2-SVP basis set are presented with p- and d-orbitals. That brings a sufficiently large number of primitive Gaussians and spherical harmonics to the numerical representation of the quantum system. A richer basis causes more complex interatomic interactions inside the molecule and a more complex structure of Hamiltonian. Also, our dataset includes notably larger molecules than the ones presented in the MD17 or QM9 dataset. This difference between the two datasets affects the results reported in the original paper and within our benchmark.
>
> However, we do not exclude the possibility of obtaining better results with QHNet on $\nabla^2$DFT , with minor changes in the architecture and intensive hyperparameter tuning. In our experimental setup, we do not modify neural networks more than necessary for data processing and use original hyperparameters aiming to repeat neural network setup and hyperparameters as close as possible to the original study.
>
> - Question 2
>
> > On the Github the download link for the full dataset is available for the energy dataset but not for the Hamiltonian dataset. Does the author plan to release the full Hamiltonian dataset?
>
> At this moment only tiny, small, medium, large and test splits of the Hamiltonian dataset are released. We plan to release the full Hamiltonian dataset till the end of August. Moreover, all the raw data (wavefunction archives) is already released, so anyone can compose the full dataset on their own.
>
> - Question 3
>
> > Compared to the Fock matrix, the overlap matrix can be quite easily computed with quantum chemistry software. Is it necessary to train ML models to predict the overlap matrix?
>
> We agree that the overlap matrix is easily computable when compared to the Fock one. We followed SchNOrb and PhiSNet papers, where an overlap matrix is predicted along with a Hamiltonian; however, the importance of this component is not discussed there. The QHNet model training was run without the prediction of overlap matrices, so, in principle, it is unnecessary. We hope that more experiments with different models and targets will be conducted in the future. We hypothesize that the prediction of the overlap matrix may serve as an additional training regularization.
>
> - Question 4
>
> > The results of geometry optimization in table 4 & 5 are compelling, even when the models are only trained with RDKit geometries. Do the authors observe that the optimization is good enough without finetuning on DFT trajectories?
>
> First of all, the optimization process is necessary for the validation of the models because it is the only way to get relaxed geometries. We agree that for several models, the optimization is good enough without finetuning, and we will emphasize this fact in the manuscript. Still, we see an additional performance boost after the models finetuning on the trajectories, especially in terms of the optimization success percentage metric. Please see the metrics for pretrained and finetuned PaiNN and GemNet models below.
>
> | Model | $\overline{\operatorname{pct}}_T$ | $\operatorname{pct}_{\text{success}}$ | $\operatorname{pct}_{\text{div}}$ |
> |-|-|-|-|
> | PAINN | 98.50 | 77.09 | 0.05|
> | GemNet| 100.06| 90.71 | 0 |
> | PaiNN-finetune| 99.83 | 84.35 | 0.4 |
> | GemNet-finetune | 100.005 | 94.55 | 0 |
>
> - Question 5
>
> > There appears to be some downloading issues raised in the Github page (e.g., downloading a large trunk of data in serial appears to be difficult). Does the authors plan to address these?
>
> We moved the data to a more reliable storage, so the downloading speed and stability were increased. Moreover, we will either split all the big files (larger than 100G) into chunks or provide a chunkwise-download tool.

---

> > ### Comment · Reviewer_EvHy · 2024-09-06
> >
> > Thanks for the clarifications. I think it's an interesting dataset and empirical study, and I will update my rating from 6 to 7.

---

### Author Rebuttal · Authors · 2024-08-14

We thank all the reviewers for the valuable questions and comments!

During the time after the submission, the following changes were made to the dataset and benchmark:

1. GemNet-OC, EquiformerV2, eSCN, and Graphormer3D networks were added and validated by the optimization pipeline; moreover, GemNet-OC was finetuned on the trajectories data.
2. All the data was moved to a faster and more reliable cloud storage.
3. The training and inference tutorials were updated and expanded.
4. 73% of the code was covered with tests.

All the mentioned changes will be added in the manuscript revision.

---

### Author Response · Authors · 2024-08-27

Dear Reviewers,

If there are any additional questions, clarifications, or concerns you would like us to address, we would be more than happy to discuss them. Your insights and continued engagement during this discussion phase would greatly help us in refining the paper further.

Best regards,
Authors

---

### Decision · Program_Chairs · 2024-09-26

**Decision:**

Accept (Poster)

**Comment:**

All four reviewers are positive about this work and thus an accept is recommended.